# Racial and geographic variation in effects of maternal education and neighborhood-level measures of socioeconomic status on gestational age at birth: Findings from the ECHO cohorts

**Anne L. Dunlop** [1]\*, **Alicynne Glazier Essalmi** [2], **Lyndsay Alvalos** [3], **Carrie Breton** [4], **Carlos A. Camargo** [5], **Whitney J. Cowell** [6], **Dana Dabelea** [7], **Stephen R. Dager** [8], **Cristiane Duarte** [9], **Amy Elliott** [10], **Raina Fichorova** [11], **James Gern** [12], **Monique M. Hedderson** [3], **Elizabeth Hom Thepaksorn** [13], **Kathi Huddleston** [14], **Margaret R. Karagas** [15], **Ken Kleinman** [16], **Leslie Leve** [17], **Ximin Li** [18], **Yijun Li** [18], **Augusto Litonjua** [19], **Yunin Ludena-Rodriguez** [20], **Juliette C. Madan** [21], **Julio Mateus Nino** [22], **Cynthia McEvoy** [23], **Thomas G. O'Connor** [24], **Amy M. Padula** [13], **Nigel Paneth** [2], **Frederica Perera** [9], **Sheela Sathyanarayana** [25], **Rebecca J. Schmidt** [20], **Robert T. Schultz** [26], **Jessica Snowden** [27], **Joseph B. Stanford** [28], **Leonardo Trasande** [29], **Heather E. Volk** [18], **William Wheaton** [30], **Rosalind J. Wright** [6], **Monica McGrath** [18], on behalf of program collaborators for Environmental Influences on Child Health Outcomes [¶]

1 Woodruff Health Sciences Center, School of Medicine and Nell Hodgson Woodruff School of Nursing, Emory University, Atlanta, Georgia, United States of America, 2 Department of Epidemiology and Biostatistics, College of Human Medicine, Michigan State University, East Lansing, Michigan, United States of America, 3 Division of Research, Kaiser Permanente Northern California, Oakland, California, United States of America, 4 Department of Preventive Medicine, University of Southern California, Los Angeles, California, United States of America, 5 Department of Epidemiology Harvard University, Cambridge, Massachusetts, United States of America, 6 Department of Environmental Medicine and Public Health, Icahn School of Medicine at Mount Sinai, New York, New York, United States of America, 7 Department of Epidemiology, University of Colorado Anschutz Medical Campus, Aurora, Colorado, United States of America, 8 Department of Radiology and Bioengineering, University of Washington, Seattle, Washington, United States of America, 9 Department of Psychiatry, Columbia University, New York, New York, United States of America, 10 Avera Research Institute Center for Pediatric & Community Research, Sioux Falls, South Dakota, United States of America, 11 Department of Obstetrics, Gynecology and Reproductive Biology, Brigham and Women's Hospital and Harvard Medical School, Boston, Massachusetts, United States of America, 12 Department of Medicine, University of Wisconsin, Madison, Wisconsin, United States of America, 13 Program on Reproductive Health and the Environment, Department of Obstetrics, Gynecology and Reproductive Sciences, University of California San Francisco, San Francisco, California, United States of America, 14 College of Health and Human Services, George Mason University, Fairfax, Virginia, United States of America, 15 Department of Epidemiology, Geisel School of Medicine at Dartmouth, Lebanon, New Hampshire, United States of America, 16 Department of Biostatistics and Epidemiology, University of Massachusetts Amherst, Amherst, Massachusetts, United States of America, 17 Prevention Science Institute, University of Oregon, Eugene, Oregon, United States of America, 18 Department of Biostatistics, Johns Hopkins Bloomberg School of Public Health, Baltimore, Maryland United States of America, 19 Division of Pediatric Pulmonary Medicine, Golisano Children's Hospital at Strong, University of Rochester Medical Center, Rochester, New York, United States of America, 20 Division of Environmental and Occupational Health, Public Health Sciences, School of Medicine, University of California, Davis, California, United States of America, 21 Department of Epidemiology Geisel School of Medicine at Dartmouth Hitchcock Medical Center, Hanover, New Hampshire, United States of America, 22 Obstetrics and Gynecology, Maternal and Fetal Medicine, Atrium Health, Charlotte, North Carolina, United States of America, 23 Division of Neonatal, Department of Pediatrics, Oregon Health & Science University, Portland, Oregon, United States of America, 24 Department of Psychiatry, School of Medicine and Dentistry, University of Rochester Medical Center, Rochester, New York, United States of America, 25 Department of Pediatrics, University of Washington & Seattle Children's Research Institute, Seattle, Washington, United States of America, 26 Department of Pediatrics, University of Pennsylvania Perelman School of Medicine, Philadelphia, Pennsylvania, United States of America, 27 Department of Pediatrics, University of Arkansas Medical

**Data Availability Statement:** For this disseminated meta-analysis, only metadata from the participating

cohorts was received by the ECHO Data Analysis Center (DAC); no individual-level data was received. Per ECHO policies, individuals requesting to use the metadata should contact ECHO-DAC@rti.org.

**Funding:** Please see the supporting information file S1 Funding for a complete list of all funding sources and grant numbers.

**Competing interests:** The authors have declared that no competing interests exist.

Sciences, Little Rock, Arkansas, United States of America, **28** Department of Family Preventative Medicine, University of Utah School of Medicine, Salt Lake City, Utah, United States of America, **29** Department of Pediatrics, New York University Grossman School of Medicine, New York, New York, United States of America, **30** Science and Technology Program, RTI International, Research Triangle Park, North Carolina, United States of America

¶ The program collaborators for Environmental Influences on Child Health Outcomes is provided in the Acknowledgments section.
* amlang@emory.edu

## Abstract

Preterm birth occurs at excessively high and disparate rates in the United States. In 2016, the National Institutes of Health (NIH) launched the Environmental influences on Child Health Outcomes (ECHO) program to investigate the influence of early life exposures on child health. Extant data from the ECHO cohorts provides the opportunity to examine racial and geographic variation in effects of individual- and neighborhood-level markers of socio-economic status (SES) on gestational age at birth. The objective of this study was to examine the association between individual-level (maternal education) and neighborhood-level markers of SES and gestational age at birth, stratifying by maternal race/ethnicity, and whether any such associations are modified by US geographic region. Twenty-six ECHO cohorts representing 25,526 mother-infant pairs contributed to this disseminated meta-analysis that investigated the effect of maternal prenatal level of education (high school diploma, GED, or less; some college, associate's degree, vocational or technical training [reference category]; bachelor's degree, graduate school, or professional degree) and neighborhood-level markers of SES (census tract [CT] urbanicity, percentage of black population in CT, percentage of population below the federal poverty level in CT) on gestational age at birth (categorized as preterm, early term, full term [the reference category], late, and post term) according to maternal race/ethnicity and US region. Multinomial logistic regression was used to estimate odds ratios (OR) and 95% confidence intervals (CIs). Cohort-specific results were meta-analyzed using a random effects model. For women overall, a bachelor's degree or above, compared with some college, was associated with a significantly decreased odds of preterm birth (aOR 0.72; 95% CI: 0.61–0.86), whereas a high school education or less was associated with an increased odds of early term birth (aOR 1.10, 95% CI: 1.00–1.21). When stratifying by maternal race/ethnicity, there were no significant associations between maternal education and gestational age at birth among women of racial/ethnic groups other than non-Hispanic white. Among non-Hispanic white women, a bachelor's degree or above was likewise associated with a significantly decreased odds of preterm birth (aOR 0.74 (95% CI: 0.58, 0.94) as well as a decreased odds of early term birth (aOR 0.84 (95% CI: 0.74, 0.95). The association between maternal education and gestational age at birth varied according to US region, with higher levels of maternal education associated with a significantly decreased odds of preterm birth in the Midwest and South but not in the Northeast and West. Non-Hispanic white women residing in rural compared to urban CTs had an increased odds of preterm birth; the ability to detect associations between neighborhood-level measures of SES and gestational age for other race/ethnic groups was limited due to small sample sizes within select strata. Interventions that promote higher educational attainment among women of reproductive age could contribute to a reduction in preterm

birth, particularly in the US South and Midwest. Further individual-level analyses engaging a diverse set of cohorts are needed to disentangle the complex interrelationships among maternal education, neighborhood-level factors, exposures across the life course, and gestational age at birth outcomes by maternal race/ethnicity and US geography.

## Introduction

Preterm birth (PTB) ($<$ 37 weeks' gestation) occurs at excessively high rates in the United States compared to other developed nations. Provisional data for 2018 show an overall US rate of PTB of 10.02%, up from 9.93% in 2017, which translates to 386,400 affected births annually [1, 2]. Within the United States, the rate of PTB varies considerably by race/ethnicity [1]. In 2015–2017, the highest rate of PTB was for women who were non-Hispanic black (13.6%), followed by American Indian/Alaska Native (11.3%), Hispanic (9.4%), non-Hispanic white (9.0%), and Asian/Pacific Islander (8.7%) [3]. There is also substantial state-to-state variation in rates of PTB, with the lowest rates in the northeastern and northwestern states and the highest rates in the southeastern states [3]. States with higher percentages of non-Hispanic black women have higher rates of PTB even after adjusting for state-specific rates of poverty, obesity, smoking, and teen birth [4]. There is a growing recognition of the adverse child health effects associated with early term birth (37-0/7 weeks through 38-6/7 weeks), including elevated rates of infant mortality and long-term neurological morbidity [5–7]. Both late-term (41-0/7 weeks through 41-6/7 weeks) and post-term birth (42-0/7 weeks and beyond) are associated with an increased risk for stillbirth and perinatal death; additional fetal risks of post-term births include macrosomia, neonatal seizures, and meconium aspiration [8, 9]. Limited published data have investigated risk factors for early-term, late-term, and post-term births, although a 2014 analysis demonstrated that racial disparities exist in rates of early-term delivery among US women [10].

The observed disparities in rates of gestational age at birth by race/ethnicity and geography are theorized to be related in part to differences in individual-level socioeconomic status (SES) [11]. Those of lower SES have a higher burden of a range of adverse health outcomes [12], and across various measures of individual-level SES (including maternal level of education and income, marital and employment status, and type of health insurance) there is a consistent social gradient in the risk of PTB [13]. Differences in individual-level SES do not adequately account for racial/ethnic disparities in PTB, however [14]. A substantial body of research has focused on the role of stress and racism, revealing that racial discrimination is an important source of stress for black women and that physiological responses to chronic stress contribute to their excess rates of PTB [15–19]. More recently, neighborhood-level markers of SES have been explored in relation to PTB [20–28], as they have been conceptually linked to adverse birth outcomes via pathways mediated by individual-level health behaviors, psychosocial factors, social support, stress, and access to quality health care, healthy food, and recreational facilities [29].

## Objective

In 2016, the National Institutes of Health (NIH) launched the Environmental influences on Child Health Outcomes (ECHO) program to investigate the influence of early life environmental exposures on child health and development. ECHO funds existing pediatric cohorts throughout the United States (https://www.nih.gov/echo/pediatric-cohorts) that have observed

children for many years. Participating cohorts underwent a competitive application process to become part of the ECHO consortium of cohorts, with the goal of enrolling and following more than 50,000 children from diverse racial, geographic, and sociodemographic backgrounds. The application and selection process required that cohorts commit to both integrating data already collected via extant study protocols and to collecting new data collected through the ECHO-wide cohort data collection protocol [30]. The ECHO Data Analysis Center (DAC) initiated collective analyses, an innovative adaptation of meta-analysis, as a strategy to speed early research productivity by allowing cohorts to combine extant data using extant data and to accelerate a collaborative research environment among the ECHO cohorts and components [31]. These collective analyses allowed for cohorts to combine data from across their cohort sites without having to share individual-level participant data, which was not permissible in most cases due to the extant data representing data collected under cohort-specific consent processes that did not specify sharing across cohorts or with the DAC. New data collected through the ECHO-wide cohort data collection protocol is being done under consent processes that do allow for sharing of individual-level participant data.

This collective analyses capitalizes on the extant data available from cohorts participating in the ECHO program to: 1) examine the association between both individual-level (maternal education) and neighborhood-level (census tract [CT] urbanicity, percentage of black population in CT, and percentage of population below the federal poverty level in CT) markers of SES and categories of gestational age at birth, stratifying by maternal race/ethnicity; and 2) determine whether associations between individual- and neighborhood-level markers of SES and gestational age at birth are modified by US geographic region. In addition, this study serves as a demonstration of the collective analysis process within a consortium of pediatrics when identifiable participant-level data could not be shared.

## Materials and methods

### Study population

The study population consisted of women enrolled in an ECHO cohort who delivered a live-born singleton infant for whom the following exposure and covariate information was available: maternal prenatal education and residential address or geographic region during the prenatal period for the biological mother, mother's age at delivery, and child's birth sex. Cohorts were excluded from initial consideration if their cohort-specific selection criteria restricted child enrollment based on gestational age (e.g., only PTB), or if no gestational age or maternal prenatal education information was collected. Forty-four cohorts, representing 230 recruitment sites, contributed data on 34,155 mother-infant pairs for consideration of inclusion in this analysis. Twenty-six cohorts representing 110 recruitment sites and 25,526 mother-infant pairs contributed to these analyses (Fig 1) after excluding three cohorts missing information on maternal prenatal education, one cohort lacking heterogeneity in maternal prenatal education, one cohort missing address information, four cohorts missing parity, five cohorts missing marital status, and four cohorts with small sample size (e.g., ≤ 30 participants having information on all required covariates). For each cohort that contributed data, the local institutional review board reviewed and approved the protocol for collecting data; all cohort participants provided written informed consent.

### Gestational age at birth

Gestational age at birth was categorized as defined by the American College of Obstetricians and Gynecologists: preterm (22-0/7 weeks through 36-6/7 weeks), early-term (37-0/7 weeks through 38-6/7 weeks), full-term (39-0/7 weeks through 40-6/7 weeks, the reference category),

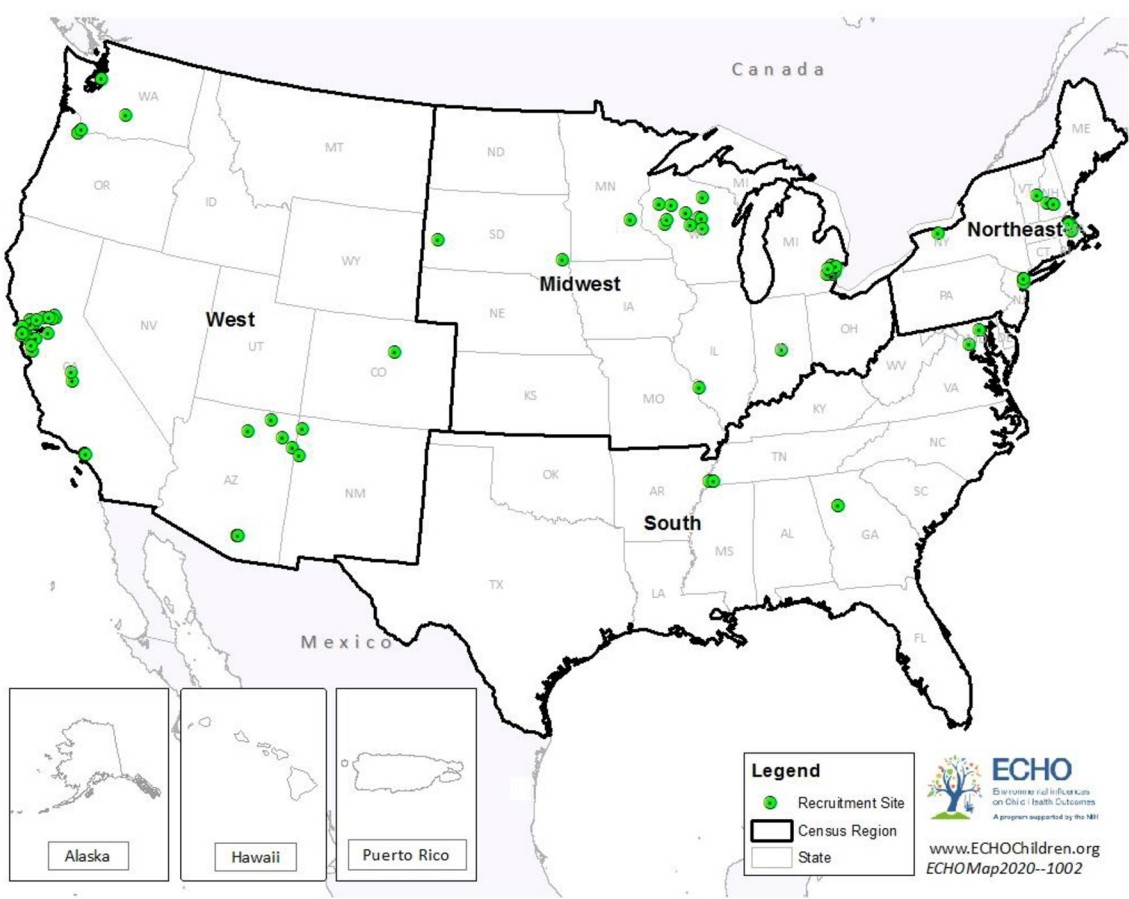

**Fig 1. Cohort recruitment sites (n = 110) by US census region.**

late-term (41-0/7 weeks through 41-6/7 weeks), and post-term (42-0/7 weeks and beyond) [32]. Late and post-term births were collapsed due to small sample sizes. Gestational age was captured according to an accepted hierarchy [33, 34] and was ascertained using the following methods: crown rump length of an early ultrasound or dating based on *in vitro* fertilization (20.7%), ultrasound taken in the second trimester with fetal biparietal diameter dating within 2 weeks of sure last menstrual period (LMP) (7.9%); ultrasound taken in the second trimester with unsure or no LMP date (1.3%); report from obstetrical medical record reporting "consensus" estimated date of delivery (55.5%); dating by LMP with a third trimester ultrasound dating within 2 weeks of estimated date of delivery by LMP dating (0.03%); dating from LMP date only (5.5%); reported by mother (8.8%). Only 0.43% of participants were missing method of ascertainment of gestational age. Within participating cohorts, individuals were excluded if gestational age was missing (n = 500) or if gestational age at birth was <22 weeks or >44 weeks (n = 10).

## Maternal level of education

Maternal prenatal level of education, by self-report, served as the individual-level measure of SES and was categorized as high school diploma, GED, or less; some college, associate's degree, vocational or technical training (the reference category); and bachelor's degree, graduate school, or professional degree.

We used the categorical variable of maternal level of education as the main individual-level measure of SES based on existing research among US populations that support that education is the dimension of SES that most strongly and consistently predicts health, especially for women and children [35, 36]. Also, maternal level of education as a categorical variable was more available across the ECHO cohort extant data. Paternal level of education was less often collected by the cohorts or, when collected, was oftentimes missing. Similarly, household income was less available across cohorts in that either entire cohorts did not collect this data or the variable response was missing within cohorts. Also, in order to be meaningfully interpreted income in relation to the federal poverty line, household income must be combined with household density, which was also not uniformly available across cohorts and/or was missing within cohort extant data. Furthermore, even when using the federal poverty thresholds, the meaning of the various income groupings varies a great deal across states and according to rural/urban status.

## Neighborhood-level SES

**Geocoding.** Primary residential address of the biological mother during pregnancy was ascertained from the earliest address collection during pregnancy. Decentralized Geomarker Assessment for Multi-Site Studies (DeGAUSS) was used to geocode the prenatal residential address for each cohort locally. Information regarding the DeGAUSS software has been described elsewhere [37]. DeGAUSS, a freely available distributed geocoding system, facilitates reproducible geocoding and geomarker assessment in multi-site studies while maintaining the confidentiality of protected health information by enabling the sites access to a single embedded geocoding engine that uses the same nationwide street address data base (the Census Bureau's Topologically Integrated Graphical Encoding and Referencing, TIGER, data) and matching code without having to share address data. The DeGAUSS geocoder used current (at the time of the study) TIGER street centerline address range files to convert residential addresses into geographical coordinates and provided assessment of qualitative precision for the geocodes [37]. Using the latest TIGER street centerline address range files was appropriate because of improvements to street and address ranges made by census bureau over time. After geocoding, the resulting latitude/longitude coordinates were joined to the TIGER 2000 Census Tract boundaries. Since census tracts are revised only every ten years, the TIGER 2000 Census Tract boundaries were needed in order to correctly match the census tract for geocoded addresses to the 2005–2009 American Community Survey (ACS) 5-year estimates, which used the same census tract identifiers as the 2000 TIGER census tract layer. We chose the ACS 2005–2009 dataset because for births prior to 2005–2009 there is not a corresponding five-year ACS file that contains estimates for every CT (2005–2009 is the first available release of this data) and, during the analysis planning stage, 2008 seemed to be the approximate mid-point of participant enrollment. Geocodes of addresses that matched to street segments or street centers were included. There were 166 participants with missing addresses. Addresses that matched to intersections of streets, zip code centroids, or cities and addresses that failed to map to a valid latitude and longitude were excluded (n = 870). The percentage of text match between the residential address and the geocoded result was calculated for each observation. Records with less than 60% text match were excluded (n = 1071). For cohorts that did not wish to use DeGAUSS, the ECHO Data Analysis Center (DAC) offered options to supply 1) high-quality GPS-based latitude/longitude coordinates or 2) latitude/longitude coordinates generated by commercial or other high-quality geocoders (e.g., ArcGIS) that had been manually examined and corrected. The Federal Information Processing Standards (FIPS) codes were then determined based on valid geocodes using the DeGAUSS geocoder as CT identifiers. A total of 23,419 participants from 26 cohorts provided addresses that were successfully geocoded (92% of the

address records). Twenty-three cohorts geocoded maternal prenatal address using DeGAUSS; two cohorts used ArcGIS, and one cohort provided high-quality GPS-based latitude/longitude coordinates.

**Neighborhood-level SES variables.** CT variables are aggregated individual characteristics within a CT and are used to represent the neighborhood sociodemographic environment. For participants whose prenatal residential address was successfully geocoded (n = 23,419), their neighborhood-level characteristics were obtained by linking the FIPS codes to the 2005–2009 ACS 5-Year Estimate [38] for three CT level markers of SES: urbanicity (rural vs. urban [reference category]), percentage black population in CT (as a measure of segregation), and percentage population living below the 100% federal poverty line in CT (as a measure of poverty) [39]. The continuous measures of percent black population and percent population living below the federal poverty line in the CTs were dichotomized as "below national average" and "above or equal to national average," where the national averages were defined as the median percent of the tract population among all US CTs for each variable of interest. The neighborhood-level variables were coded as missing for those with missing geocoded addresses; those with missing geocoded addresses still participated in descriptive analyses overall and by race/ethnicity but were excluded from adjusted analyses. The number of CTs per cohort ranges from 32 to 592, with an average of 257 CTs per cohort. The mean number of observations per CT ranges from 1 to 9.7, with an average of 3.7.

We selected the three CT level markers of SES based on past research that investigated the relationship between census tract markers of SES and preterm birth, which found that the single-measure variable of percentage of persons below the poverty level performed as well as more complex, composite measures of economic deprivation [23] in conjunction with our a priori interest in exploring CT markers of rural/urban status and segregation in conjunction with poverty. Specifically, Carmichael et al. [23] found that the single-measure of percent of the CT population below the poverty level correlated well with an eight-measure index of CT variables (r = 0.86) and resulted in the same conclusions. We chose to consider SES measures at the level of CT based on literature showing that CT-level analyses resulted in maximal geocoding linkage (i.e., the highest proportion of recorded geocoded and linked to census-defined geography) and that measures of economic deprivation at the CT-level were most sensitive to expected socioeconomic gradients in health among non-Hispanic black, non-Hispanic white, and Hispanic men and women [40, 41]. We used five-year data releases from ACS as only the five-year releases are guaranteed to have estimates for every CT in the United States, with the 2005–2009 five-year release being the first release available. In this disseminated meta-analysis, addresses were geocoded at the cohort level using DeGAUSS as participant address could not be shared with the DAC; disseminating analytical code to the cohorts to run locally using time-weighted averages for each woman was not possible across all of the participating cohorts.

## Race/ethnicity

Maternal race and ethnicity were categorized as non-Hispanic white, non-Hispanic black, non-Hispanic other, and Hispanic. Non-Hispanic other included non-Hispanic persons who identified their race as American Indian or Alaska Native, Native Hawaiian or other Pacific Islander, Asian, multiple race, or other race.

## Geographic region

Geographic region was assigned based on the US census regions (Fig 1)—categorized as West, Midwest, South, or Northeast—for the state in which the mother resided during the prenatal period.

## Covariates

Covariates were selected based on their association with gestational age at birth in the litera-
ture [42, 43] and availability within the ECHO cohorts [44]. Covariates included in all statisti-
cal models were maternal age in years (as a continuous variable), maternal parity (number of
births after 20 weeks' gestation excluding the index child, classified as nulliparous, 1–2 births,
or 3 or more births), marital status (married or single/widowed/separated/divorced), and child
sex at birth (male or female).

Additional covariates related to maternal health conditions or behaviors (defined according
to a standardized data dictionary; S1 Table) were included in subsequent models and evaluated
for their effect on the measures of association between maternal prenatal education and neigh-
borhood-level measures of SES and gestational age at birth outcomes. These additional covari-
ates included maternal prenatal body mass index (calculated from maternal height and weight
during pregnancy at the earliest prenatal time point available; categorized as underweight,
healthy weight, overweight, or obese according to accepted definitions) [45] along with the
presence or absence (yes/no) of the following diagnoses before or during the index pregnancy:
type 1 or type 2 diabetes mellitus; chronic hypertension; chronic infections (including HIV
and hepatitis B or C); other chronic health conditions including asthma, other lung disease,
cardiac disease other than hypertension or cardiovascular disease, renal disease, and thyroid
disease; prenatal tobacco use, alcohol use, and use of marijuana, stimulants, opiates (other
than during labor); pregnancy complications during the index pregnancy including premature
rupture of membranes, gestational hypertension, preeclampsia, gestational diabetes, intrauter-
ine growth restriction, placental abruption and previa; urinary tract infection including
asymptomatic bacteriuria, cystitis or lower tract infection, pyelonephritis or upper tract infec-
tion; reproductive tract infections such as bacterial vaginosis, chlamydia, gonorrhea, trichomo-
niasis, cervicitis, and pelvic inflammatory disease; receipt of prenatal care (yes/no) and
prenatal health insurance (private/not private). The number of cohorts with information var-
ied for each covariate.

## Data analysis

Existing data from eligible ECHO cohorts were utilized to conduct a disseminated meta-
analysis using results from standardized cohort-specific analyses [31]. The ECHO DAC pro-
vided cohorts with a data dictionary with variable names and definitions, which cohorts used
to create a standardized data set, and a common statistical code, which cohorts used to per-
form a standardized analysis upon their cohort-specific dataset. This approach standardizes
the statistical methods as individual cohorts conduct the same analyses. The code was
designed to automatically save all estimates and variance-covariance matrices needed to per-
form the meta-analysis. The code also included quality control measures. Any errors in the
cohort-specific dataset stopped the program and flagged errors to the cohort. Errors needed
to be rectified for the program to successfully run. The ECHO DAC received and reviewed
cohort-specific metadata and descriptive statistics, including information about any data dis-
crepancies and worked with the cohort to resolve any errors, to create the data set for dissem-
inated meta-analysis.

**Multiple imputation.** Data were imputed by multiple imputation through chained equa-
tions with the assumption that values were missing at random at the cohort level based on
cohort covariate availability with 10 imputations [46]. Predictive mean matching [47–50],
logistic regression [47, 51–53], and polytomous regression [47, 51–53] were used to impute
continuous, binary, and categorical variables, respectively. If a covariate was missing more
than 50% of its values, or if the variable was a composite variable summarizing other covariates,

data were not imputed. CT information was imputed as continuous variables then dichotomized for analyses.

**Statistical modeling.** Multinomial logistic regression was used to estimate odds ratios (OR) and 95% confidence intervals (CIs) for preterm, early-term, and late-post-term birth (reference category: full-term birth) in relation to the individual-level (maternal education) and neighborhood-level (urbanicity, percent black, percent population below 100% federal poverty line) measures of SES, adjusting for maternal age, parity, marital status, and child sex. Models were stratified by race/ethnicity (non-Hispanic white, non-Hispanic black, non-Hispanic of any other race, or Hispanic of any race) and region (Northeast, South, Midwest, or West).

Additional covariates related to maternal behaviors and health conditions were added to the models sequentially in sets (keeping each previous set), in the order as they are presented above, and individually for each set, and the effect of the addition of these variables on the associations between the individual and neighborhood-level measures of SES and gestational age at birth was assessed.

All statistical models were performed on non-imputed data and imputed data. A conservative approach was taken in that regressions were only run if there were at least 30 complete observations in each cohort. We pooled the exposure estimates from cohorts with at least five observations in both the reference category and the corresponding level of interest for the outcome and exposure. Therefore, the number of cohorts contributing to each association varied based on these sample size constraints. We employed this restriction to address concern regarding unstable estimates due to small sample sizes. Results were similar between non-imputed and imputed data; therefore, final analyses were based on the imputed data since parameter estimates were more stable.

Cohort-specific results were meta-analyzed using a random effects model [54]. All analyses were performed with the use of R software [55]. The *mice* package [47] was used for multiple imputation, and the *nnet* package [51] was used for multinomial logistic regression. Random effect meta-analyses [56] were performed using R-package *meta* [57]. Forest plots were generated using the R-package *forestplot 1.7.2* [51]. The Q test and $I^2$ statistics were used to measure the heterogeneity across the cohorts.

## Results

### Characteristics of the study population overall and by gestational age at birth

The distribution of characteristics overall and according to gestational age at birth for included cohorts is presented in Table 1. Of the 25,526 mother-infant pairs, 14,813 (58.0%) delivered full-term; 1,864 delivered preterm (7.3%), 6,191 delivered early-term (24.3%), and 2,658 (10.4%) delivered late- or post-term. The largest percentage of participating women were non-Hispanic white (48.7%), with a bachelor's degree or above (42.7%), and were married (69.8%). The mean maternal age at child's birth was 30 years old. Most of the women (72.2%) reported receiving prenatal care. Tobacco and alcohol were used by 10.8% and 19.1% of the women during pregnancy, respectively. Most women were parous with 1–2 (47%) or ≥3 (9.1%) prior births; only 2.9% of the women had a history of a prior PTB. In the index pregnancy, 12.8% experienced a cardiometabolic complication of pregnancy, 8.8% an obstetrical complication, 4.1% a prenatal urinary tract infection, and 5.1% a reproductive tract infection. The largest percentage of women lived in the West (33.9%), followed by the Northeast (25.8%), Midwest (22.2%), and the South (18.0%). Fifty-one percent of pregnant women lived where the percentage of the CT population that was black was below the national average of 3.6% (based on the

**Table 1. Descriptive characteristics of mothers of singleton live births by gestational age at birth category.**

| Variable | Total sample | Preterm birth | Early-term birth | Full-term birth | Late- or post-term birth |
|---|---|---|---|---|---|
| | N = 25526 (column %) | (22–<37 weeks) | (37–<39 weeks) | (39–<41 weeks) | (41–43 weeks) |
| | | n = 1864 (7.3%) | n = 6191 (24.3%) | n = 14,813 (58.0%) | n = 2658 (10.4%) |
| Child's birth year | | | | | |
| 1990 or earlier | 1653 (6.5%) | 21 (1.3%) | 238 (14.4%) | 1183 (71.6%) | 211 (12.8%) |
| 1991–2000 | 1632 (6.4%) | 104 (6.4%) | 367 (22.5%) | 890 (54.5%) | 271 (16.6%) |
| 2001–2010 | 6885 (27.0%) | 554 (8.1%) | 1741 (25.3%) | 3850 (55.9%) | 740 (10.8%) |
| 2011–2017 | 15309 (60.0%) | 1179 (7.7%) | 3830 (25%) | 8865 (57.9%) | 1435 (9.4%) |
| Missing | 47 (0.2%) | 6 (12.8%) | 15 (31.9%) | 25 (53.2%) | 1 (2.1%) |
| Prenatal U.S. region | | | | | |
| Northeast | 6588 (25.8%) | 540 (8.2%) | 1523 (23.1%) | 3629 (55.1%) | 896 (13.6%) |
| South | 4601 (18.0%) | 385 (8.4%) | 1197 (26%) | 2785 (60.5%) | 234 (5.1%) |
| Midwest | 5681 (22.2%) | 402 (7.1%) | 1482 (26.1%) | 3203 (56.4%) | 594 (10.5%) |
| West | 8655 (33.9%) | 537 (6.2%) | 1989 (23%) | 5195 (60%) | 934 (10.8%) |
| Missing | 1 (0%) | 0 (0%) | 0 (0%) | 1 (0%) | 0 (0%) |
| Child sex | | | | | |
| Male | 12965 (50.8%) | 1029 (7.9%) | 3229 (24.9%) | 7389 (57%) | 1318 (10.2%) |
| Female | 12536 (49.1%) | 832 (6.6%) | 2956 (23.6%) | 7408 (59.1%) | 1340 (10.7%) |
| Missing | 25 (0.1%) | 3 (12%) | 6 (24%) | 16 (64%) | 0 (0%) |
| Maternal age at child's birth | | | | | |
| Mean (min, max) | 30 (14, 50) | 30 (16, 47) | 30 (14, 49) | 30 (14, 50) | 30 (16, 45) |
| Missing | 54 | 5 (0.2%) | | 31 | 2 |
| Maternal marital status | | | | | |
| Married | 17810 (69.8%) | 1170 (6.6%) | 4159 (23.3%) | 10554 (59.3%) | 1927 (10.8%) |
| Single/widowed/separated/divorced | 6151 (24.1%) | 551 (9%) | 1622 (26.4%) | 3391 (55.1%) | 587 (9.5%) |
| Missing | 1565 (6.1%) | 143 (9%) | 410 (26.2%) | 868 (55.5%) | 144 (9.2%) |
| Maternal race/ethnicity | | | | | |
| White, Non-Hispanic | 12425 (48.7%) | 715 (5.8%) | 2747 (22.1%) | 7406 (59.6%) | 1557 (12.5%) |
| Black, Non-Hispanic | 3664 (14.4%) | 383 (10.5%) | 984 (26.9%) | 1999 (54.6%) | 298 (8.1%) |
| Non-Hispanic any race | 3846 (15.1%) | 329 (8.6%) | 1069 (27.8%) | 2131 (55.4%) | 317 (8.2%) |
| Hispanic of any race | 4813 (18.9%) | 385 (8%) | 1186 (24.6%) | 2804 (58.3%) | 438 (9.1%) |
| Missing | 778 (3.0%) | 52 (6.7%) | 205 (26.4%) | 473 (60.8%) | 48 (6.2%) |
| Maternal prenatal education | | | | | |
| High school or less | 6713 (26.3%) | 590 (8.8%) | 1767 (26.3%) | 3730 (55.6%) | 626 (9.3%) |
| Some college | 6162 (24.1%) | 466 (7.6%) | 1503 (24.4%) | 3591 (58.3%) | 602 (9.8%) |
| Bachelor's or above | 10889 (42.7%) | 670 (6.2%) | 2444 (22.4%) | 6494 (59.6%) | 1281 (11.8%) |
| Missing | 1762 (6.9%) | 138 (7.8%) | 477 (27.1%) | 998 (56.6%) | 149 (8.5%) |
| Percentage of census tract population that is black | | | | | |
| Below national average | 10884 (42.6%) | 752 (6.9%) | 2575 (23.7%) | 6316 (58%) | 1241 (11.4%) |
| Above national average | 12535 (49.1%) | 979 (7.8%) | 3114 (24.8%) | 7283 (58.1%) | 1159 (9.3%) |
| Missing | 2107 (8.3%) | 133 (6.3%) | 502 (23.8%) | 1214 (57.6%) | 258 (12.2%) |
| Percentage of tract population that is below the 100% federal poverty line | | | | | |
| Below national average | 13796 (54.0%) | 948 (6.9%) | 3253 (23.6%) | 8151 (59.1%) | 1444 (10.5%) |
| Above national average | 9622 (37.7%) | 782 (8.1%) | 2436 (25.3%) | 5448 (56.6%) | 956 (9.9%) |
| Missing | 2108 (8.3%) | 134 (6.4%) | 502 (23.8%) | 1214 (57.6%) | 258 (12.2%) |

*(Continued)*

**Table 1.** (Continued)

| Variable | Total sample | Preterm birth | Early-term birth | Full-term birth | Late- or post-term birth |
|---|---|---|---|---|---|
| | N = 25526 (column %) | (22–<37 weeks) | (37–<39 weeks) | (39–<41 weeks) | (41–43 weeks) |
| | | n = 1864 (7.3%) | n = 6191 (24.3%) | n = 14,813 (58.0%) | n = 2658 (10.4%) |
| Urbanicity of prenatal address | | | | | |
| Urban | 17884 (70.1%) | 1320 (7.4%) | 4291 (24%) | 10424 (58.3%) | 1849 (10.3%) |
| Rural | 5535 (21.7%) | 411 (7.4%) | 1398 (25.3%) | 3175 (57.4%) | 551 (10%) |
| Missing | 2107 (8.3%) | 133 (6.3%) | 502 (23.8%) | 1214 (57.6%) | 258 (12.2%) |
| Prenatal maternal tobacco smoking | | | | | |
| Yes | 2766 (10.8%) | 245 (8.9%) | 751 (27.2%) | 1505 (54.4%) | 265 (9.6%) |
| No | 21306 (83.5%) | 1476 (7%) | 5046 (23.7%) | 12516 (58.7%) | 2268 (10.6%) |
| Missing | 1454 (5.7%) | 143 (9.8%) | 394 (27.1%) | 792 (54.5%) | 125 (8.6%) |
| Prenatal maternal alcohol use | | | | | |
| Yes | 4876 (19.1%) | 350 (7.2%) | 1107 (22.7%) | 2793 (57.3%) | 626 (12.8% |
| No | 15916 (62.4%) | 1247 (7.8%) | 4111 (25.8%) | 9096 (57.2%) | 1462 (9.2%) |
| Missing | 4734 (18.5%) | 267 (5.6%) | 973 (20.6%) | 2924 (61.8%) | 570 (12%) |
| Receipt of prenatal care | | | | | |
| No | 28 (0.1%) | 3 (10.7%) | 6 (21.4%) | 17 (60.7%) | 2 (7.1%) |
| Yes | 18435 (72.2%) | 1468 (8%) | 4620 (25.1%) | 10487 (56.9%) | 1860 (10.1%) |
| Missing | 7063 (27.7%) | 393 (5.6%) | 1565 (22.2%) | 4309 (61%) | 796 (11.3%) |
| Urinary tract infections | | | | | |
| Yes | 1046 (4.1%) | 112 (10.7%) | 293 (28.0%) | 562 (53.7%) | 79 (7.6%) |
| No | 10843 (42.5%) | 842 (7.8%) | 2791 (25.7%) | 6216 (57.3%) | 994 (9.2%) |
| Missing | 13637 (53.4%) | 910 (6.7%) | 3107 (22.8%) | 8035 (58.9%) | 1585 (11.6%) |
| Reproductive tract infections | | | | | |
| Yes | 1306 (5.1%) | 124 (9.5%) | 371 (28.4%) | 711 (54.4%) | 100 (7.7%) |
| No | 8411 (33.0%) | 661 (7.9%) | 2200 (26.2%) | 4739 (56.3%) | 811 (9.6%) |
| Missing | 15809 (61.9%) | 1079 (6.8%) | 3620 (22.9%) | 9363 (59.2%) | 1747 (11.1%) |
| Parous | | | | | |
| Nulliparous | 10417 (40.8%) | 795 (7.6%) | 2301 (22.1%) | 5882 (56.5%) | 1439 (13.8%) |
| 1–2 children | 11999 (47.0%) | 788 (6.6%) | 3016 (25.1%) | 7232 (60.3%) | 963 (8%) |
| 3 or more children | 2313 (9.1%) | 231 (10%) | 676 (29.2%) | 1233 (53.3%) | 173 (7.5%) |
| Missing | 797 (3.1%) | 50 (6.3%) | 198 (24.8%) | 466 (58.5%) | 83 (10.4%) |
| Maternal body mass index | | | | | |
| Mean (min, max) | 27 (12, 74) | 28 (12, 73) | 27 (14, 74) | 27 (14, 66) | 27 (16, 60) |
| Missing | 4492 (17.6%) | 252 | 961 | 2755 | 524 |
| History of preterm birth | | | | | |
| Nulliparous | 10417 (40.8%) | 795 (7.6%) | 2301 (22.1%) | 5882 (56.5%) | 1439 (13.8%) |
| No | 6916 (27.1%) | 424 (6.1%) | 1874 (27.1%) | 4127 (59.7%) | 491 (7.1%) |
| Yes | 740 (2.9%) | 161 (21.8%) | 264 (35.7%) | 294 (39.7%) | 21 (2.8%) |
| Missing | 7453 (29.2%) | 484 (6.5%) | 1752 (23.5%) | 4510 (60.5%) | 707 (9.5%) |
| Any maternal prenatal cardiometabolic complications[1] | | | | | |
| Yes | 3256 (12.8%) | 469 (14.4%) | 1087 (33.4%) | 1528 (46.9%) | 172 (5.3%) |
| No | 12407 (48.6%) | 753 (6.1%) | 2904 (23.4%) | 7216 (58.2%) | 1534 (12.4%) |
| No Known | 4686 (18.4%) | 318 (6.8%) | 1161 (24.8%) | 2887 (61.6%) | 320 (6.8%) |
| Missing | 5177 (20.3%) | 324 (6.3%) | 1039 (20.1%) | 3182 (61.5%) | 632 (12.2%) |
| Any maternal obstetrical complications[2] | | | | | |

*(Continued)*

**Table 1.** (Continued)

| Variable | Total sample | Preterm birth | Early-term birth | Full-term birth | Late- or post-term birth |
|---|---|---|---|---|---|
| | N = 25526 (column %) | (22–<37 weeks) | (37–<39 weeks) | (39–<41 weeks) | (41–43 weeks) |
| | | n = 1864 (7.3%) | n = 6191 (24.3%) | n = 14,813 (58.0%) | n = 2658 (10.4%) |
| Yes | 2247 (8.8%) | 438 (19.5%) | 672 (29.9%) | 986 (43.9%) | 151 (6.7%) |
| No | 5891 (23.1%) | 331 (5.6%) | 1439 (24.4%) | 3611 (61.3%) | 510 (8.7%) |
| No known | 9458 (37.1%) | 605 (6.4%) | 2478 (26.2%) | 5452 (57.6%) | 923 (9.8%) |
| Missing | 7930 (31.1%) | 490 (6.2%) | 1602 (20.2%) | 4764 (60.1%) | 1074 (13.5%) |

[1] Any maternal prenatal cardiometabolic complications was defined as pre-eclampsia, gestational hypertension, gestational diabetes mellitus in the index pregnancy. 'Yes' indicates a participant for whom the cohort had documentation of a maternal prenatal cardiometabolic complication; 'no' indicates a participant for whom the cohort had documentation of not having any maternal prenatal cardiometabolic complications; 'no known' indicates a participant for whom the cohort had some documentation of not having a maternal prenatal cardiometabolic complications yet did not have complete information on pre-eclampsia, gestational hypertension, and gestational diabetes mellitus in the index pregnancy.

[2] Any maternal obstetrical complications was defined as intrauterine growth restriction (IUGR), premature rupture of membranes (PROM), placental abruption, and/or placenta previa in the index pregnancy. 'Yes' indicates a participant for whom the cohort had documentation of a maternal obstetrical complications; 'no' indicates a participant for whom the cohort had documentation of not having maternal obstetrical complications; 'no known' indicates a participant for whom the cohort had some documentation of not having a maternal obstetrical complications yet did not have complete information on IUGR, PROM, placental abruption, and/or placenta previa in the index pregnancy.

Except where indicated, data shown are n (%). Values may not sum to 100% because of rounding.

median of the county-level characteristic from the 2005–2009 ACS), and 54% of the pregnant women lived in a CT where the percentage of the tract population that was below 100% of the federal poverty line was below the national average of 11.5% (based on the median of the county-level characteristic from the 2005–2009 ACS) [38]. The majority of pregnant women (70.1%) lived in an urban environment.

## Association between race/ethnicity and gestational age at birth

In unadjusted analyses, there were significant differences in the odds of preterm, early-term, and late-post-term relative to full-term birth according to maternal race/ethnicity. Compared to non-Hispanic white women, women of all other races/ethnicities had a significantly increased odds of preterm or early-term birth (Table 2). Non-Hispanic black women had the highest odds of PTB (crude odds ratio [cOR] 2.00, 95% CI: 1.65–2.42) and early-term birth (cOR 1.36, 95% CI: 1.14–1.61) compared to Non-Hispanic white women. Regarding late- or post-term birth, non-Hispanic black women (cOR 0.76, 95% CI: 0.61–0.95) and non-Hispanic women of other race/ethnicity (cOR 0.84, 9% CI: 0.72–0.98) had significantly decreased odds compared to non-Hispanic white women. A detailed breakdown of the race and ethnicity frequencies of mothers included in the study by gestational age at birth category is available in S2 Table.

## Association between education and gestational age at birth overall and by race/ethnicity

For women overall, a bachelor's degree or above, compared with some college, was associated with a significantly decreased odds of PTB (adjusted odds ratio [aOR] 0.72, 95% CI: 0.61–0.86), whereas a high school education or less was associated with an increased odds of early-

**Table 2. Association of maternal race/ethnicity and gestational age at birth categories[1].**

| Maternal Race/Ethnicity | Preterm birth | Early term birth | Late or post-term birth |
| --- | --- | --- | --- |
| | (22–<37 weeks) | (37–<39 weeks) | (41–43 weeks) |
| | cOR (95% CI) | cOR (95% CI) | cOR (95% CI) |
| Non-Hispanic white | 1 (ref) | 1 (ref) | 1 (ref) |
| Non-Hispanic black | 2.00 (1.65–2.42) | 1.36 (1.14–1.61) | 0.76 (0.61–0.95) |
| Non-Hispanic other race[2] | 1.61 (1.27–2.05) | 1.25 (1.13–1.39) | 0.84 (0.72–0.98) |
| Hispanic | 1.28 (1.06–1.54) | 1.25 (1.09–1.43) | 0.91 (0.75–1.1) |

Abbreviations: CI = confidence interval; cOR = crude odds ratio; ref = reference.

[1] Unadjusted multinomial regression. Reference category is full-term births defined as gestational age at birth of ≤39–<41 weeks.

[2] Non-Hispanic other race included non-Hispanic persons who identified their race as American Indian or Alaska Native, Native Hawaiian or other Pacific Islander, Asian, multiple race, or other race.

term birth (aOR 1.10, 95% CI: 1.00–1.21) compared to full-term birth, adjusting for CT variables as well as maternal age, parity, marital status, and child sex (Table 3).

Among non-Hispanic white women, a bachelor's degree or above was also associated with a significantly decreased odds of PTB (aOR 0.74, 95% CI: 0.58–0.94) as well as early-term birth (aOR 0.84, 95% CI: 0.74–0.95). There were no significant associations between maternal education and gestational age at birth category among women of racial/ethnic groups other than non-Hispanic white (Table 3). Forest plots showed no significant heterogeneity across the cohorts. The inclusion of additional covariates related to maternal behaviors and health conditions did not meaningfully affect the associations between prenatal maternal education and gestational age at delivery (S3 Table).

## Association between education and gestational age at birth by race/ethnicity and US region

The adjusted association between maternal education and gestational age at birth was found to vary according to US region (Fig 2). For the Northeast and West, maternal level of education was not significantly associated with gestational age at birth for women overall or according to race/ethnicity. However, for the Midwest and South, a bachelor's degree or above was associated with a significantly decreased odds of delivering PTB for women overall.

## Association between neighborhood-level factors and gestational age at birth by race/ethnicity and US region

There were no significant associations between CT urbanicity, percent black in CT, or percent poverty in CT and gestational age at birth overall or for women of most races/ethnicities (Table 3), adjusting for maternal education, maternal age, parity, marital status, and child sex. This analysis was limited in its ability to investigate the effect of the CT variables and gestational age categories due to small sample sizes for the various racial/ethnic groupings; the distribution of mothers according to race/ethnicity and the neighborhood-level markers of SES by gestational age at birth category are given in S4 Table. Among non-Hispanic white women, prenatal residence in a rural compared to urban area was associated with an increased odds of delivering preterm (aOR 1.45, 95% CI: 1.09–1.92) compared to delivering full-term. Furthermore, the inclusion of additional covariates related to maternal health conditions and behaviors did not meaningfully affect the associations between these neighborhood-level measures

ONE Racial and geographic variation in effects of maternal socioeconomic status on gestational age at birth

**Table 3. Association between maternal education, neighborhood-level socioeconomic status, and gestational age at birth by maternal race/ethnicity.**

| | Preterm | | | | | Early-Term | | | | | Late- or Post-Term | | | | |
|---|---|---|---|---|---|---|---|---|---|---|---|---|---|---|---|
| | Adjusted OR (95% CI) | | | | | Adjusted OR (95% CI) | | | | | Adjusted OR (95% CI) | | | | |
| | Overall | NH white | NH black | NH other race | Hispanic | Overall | NH white | NH black | NH other race | Hispanic | Overall | NH white | NH black | NH other race | Hispanic |
| **Prenatal Maternal Education[1]** | | | | | | | | | | | | | | | |
| Bachelor's or above | 0.72 (0.61–0.86) | 0.74 (0.58–0.94) | 0.85 (0.38–1.93) | 0.63 (0.35–1.15) | 0.63 (0.29–1.34) | 0.92 (0.85–1.01) | 0.84 (0.74–0.95) | 0.95 (0.73–1.24) | 1.05 (0.81–1.36) | 1.14 (0.82–1.58) | 1.02 (0.89–1.17) | 1.00 (0.85–1.18) | 1.05 (0.52–2.11) | 0.98 (0.63–1.50) | 1.93 (0.91–4.08) |
| Some college | ref | ref | ref | ref | ref | ref | ref | ref | ref | ref | ref | ref | ref | ref | ref |
| High school or less | 1.14 (0.98–1.33) | 1.41 (0.96–2.08) | 1.28 (0.93–1.75) | 1.15 (0.77–1.72) | 0.83 (0.53–1.28) | 1.11 (1.00–1.21) | 1.07 (0.90–1.27) | 1.26 (0.96–1.66) | 1.14 (0.89–1.47) | 1.07 (0.85–1.34) | 1.05 (0.91–1.21) | 1.02 (0.81–1.27) | 1.13 (0.77–1.66) | 1.24 (0.61–2.50) | 0.83 (0.52–1.33) |
| **Census tract—Urbanicity[2]** | | | | | | | | | | | | | | | |
| Rural | 1.12 (0.82–1.54) | 1.45 (1.09–1.92) | —[3] | 0.76 (0.25–2.33) | —[3] | 1.01 (0.90–1.13) | 1.03 (0.89–1.19) | 1.18 (0.60–2.30) | 1.00 (0.77–1.31) | 1.76 (0.88–3.51) | 0.95 (0.78–1.16) | 0.88 (0.71–1.09) | —[3] | 1.24 (0.76–2.02) | 1.30 (0.28–6.00) |
| Urban | ref | ref | ref | ref | ref | ref | ref | ref | ref | ref | ref | ref | ref | ref | ref |
| **Census tract—Percent Black[4]** | | | | | | | | | | | | | | | |
| % Black above | 1.09 (0.95–1.26) | 1.04 (0.83–1.30) | —[3] | 0.95 (0.61–1.48) | 1.07 (0.75–1.54) | 1.06 (0.95–1.17) | 1.04 (0.91–1.17) | 0.66 (0.40–1.08) | 1.02 (0.82–1.26) | 0.99 (0.78–1.25) | 0.96 (0.86–1.08) | 1.08 (0.92–1.27) | —[3] | 0.78 (0.53–1.13) | 0.85 (0.60–1.19) |
| % black below | ref | ref | ref | ref | ref | ref | ref | ref | ref | ref | ref | ref | ref | ref | ref |
| **Census tract—Percent Poverty[5]** | | | | | | | | | | | | | | | |
| % Poverty above | 1.04 (0.91–1.19) | 1.16 (0.90–1.49) | 0.84 (0.52–1.35) | 1.04 (0.70–1.52) | 1.10 (0.77–1.58) | 1.06 (0.98–1.15) | 1.09 (0.96–1.23) | 1.18 (0.93–1.50) | 0.87 (0.70–1.07) | 1.10 (0.89–1.35) | 1.02 (0.92–1.14) | 1.02 (0.87–1.19) | 0.90 (0.57–1.43) | 1.39 (0.98–1.96) | 1.03 (0.76–1.41) |
| % Poverty below | ref | ref | ref | ref | ref | ref | ref | ref | ref | ref | ref | ref | ref | ref | ref |

Abbreviations: CI = confidence interval; NH = Non-Hispanic; OR = odds ratio; ref = reference.

[1] Multinomial logistic regression adjusted for census tract urbanicity, census tract-percent black, census tract-percent poverty, maternal age, parity, marital status, and child sex.

[2] Multinomial logistic regression adjusted for prenatal maternal education, census tract-percent black, census tract-percent poverty, maternal age, parity, marital status, and child sex.

[3] Meta-analysis was unable to be performed due to unstable cohort-specific estimates.

[4] Multinomial logistic regression adjusted for prenatal maternal education, census tract urbanicity, census tract-percent poverty, maternal age, parity, marital status, and child sex.

[5] Multinomial logistic regression adjusted for prenatal maternal education, census tract urbanicity, census tract-percent black, maternal age, parity, marital status, and child sex.

of SES and gestational age at birth (S3 Table). The adjusted associations between the neighborhood-level SES variables and gestational age at birth did not vary according to US region (results not shown).

## Discussion

Using extant data available from the ECHO cohorts, this study found that, among women overall, level of maternal prenatal education was associated with gestational age at birth category. Specifically, for women overall, maternal education of a bachelor's degree or above, compared to some college, was associated with a significantly decreased odds of PTB whereas a

## A. Preterm birth compared to full term birth

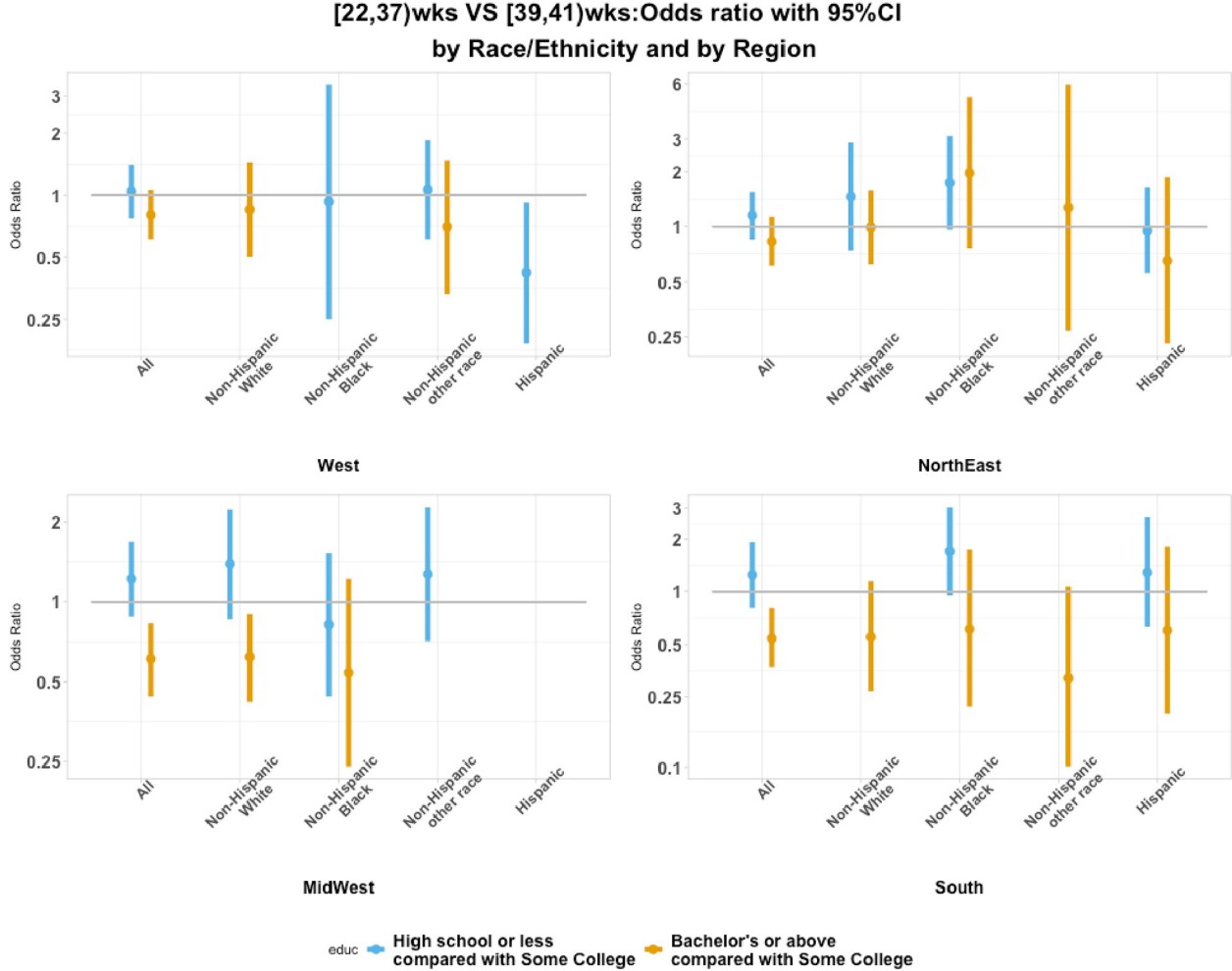

**Fig 2. Association between maternal prenatal education and gestational age at birth by maternal race/ethnicity and US region.**

high school education or less was associated with an increased odds of early-term birth compared to full-term birth. Among non-Hispanic white women, the relationship between a bachelor's degree or higher remained associated with a significantly decreased odds of PTB and, furthermore, was associated with a significantly decreased odds of early-term birth. There were no significant associations between maternal education and gestational age at birth category among women of racial/ethnic groups other than non-Hispanic white.

The relationship between higher maternal level of education and lower PTB risk has been previously documented in the United States [35, 36]. Although our results were not significant for women of all racial/ethnic groups, point estimates suggested an association between increased levels of maternal education and decreased odds of PTB. Lack of statistical power may have been due to smaller sample sizes for the race/ethnicity-specific strata. It is also possible that the effect of maternal level of education is not as related to PTB among women of race/ethnicity other than non-Hispanic white. One explanation for the persistence of racial/ethnic disparities after accounting for measures of SES, such as maternal education, is that

specified levels of SES are not equivalent across racial/ethnic groups [58, 59]. At every level of education, African American women have lower earnings and less accumulated wealth than white women and face higher average costs for housing, food, and insurance while having more people dependent on their incomes, making racial/ethnic groups not comparable at a given level of SES [60].

In this study, the association between maternal education and gestational age at birth varied according to US region, with higher levels of maternal education associated with a decreased odds of PTB in the Midwest and South but not associated with gestational age at birth in the Northeast and West. According to the US Census Bureau, the Midwest and South have the lowest percentages of individuals attaining college education [61]. Few studies have directly evaluated how the relationship between antecedent factors, such as socioeconomic and maternal health factors, vary across US geography and according to race/ethnicity [62]. Across the United States there is wide variation in policies that affect women's and children's access to and utilization of health care and other human services; reports by the Center for American Progress outline policy solutions to improve maternal and infant health outcomes and summarize policy-level information at the state-level [63, 64]. Policies related to state-level earned income tax credit laws have also been shown to affect maternal health behaviors and infant health outcomes [65]. In regard to the neighborhood-level measures of SES, non-Hispanic white women residing in a rural CT compared to an urban CT had an increased odds of PTB. The ability to detect associations between neighborhood-level measures of SES and gestational age by race/ethnicity and US region was limited due to small sample sizes within select strata. It is likely that the prevalence of factors influencing PTB risk differs for urban versus rural regions, and the magnitude of the effects of individual risk factors also may differ by urban versus rural residence; thus, it is possible that neighborhood-level measures of SES could be conflated with geographic region. Based on 2000–2005 data on live births from the US National Center for Health Statistics, the prevalence of PTB did not vary between urban versus less populated areas; however, the impact of sociodemographic factors on PTB was limited to urban areas [66]. In an investigation of traffic-related air pollution, nitrogen dioxide concentrations were associated with higher PTB rates but only in urban regions [67]. Moreover, efforts to reduce PTB may differentially impact rural pregnancies; for example, in Alabama, PTB rates in urban areas have decreased since 2006, but this trend has not been observed in rural areas [68]. Thus, future studies of diverse populations such as non-Hispanic black women, Native American and Alaskan Native women, who have the highest rates of PTB [69], will be needed to elucidate urban-rural and racial/ethnic differences, and the causes of any observed disparities have the potential to identify opportunities for targeted prevention interventions.

The lack of association between CT percent poverty and CT percent black and gestational age at birth category may be due to heterogeneity within our study population in regard to the association of these CT level measures with lack of health care, discrimination, or other factors; alternately (or in addition), the lack of association may stem from inadequate power. Despite this study's relatively large sample size, some categories remained small when we examined the interplay of multiple factors. Neighborhood-level measures of SES have been assessed to investigate disparities in PTB in previous studies with mixed findings. In one study, neighborhood deprivation, as measured by eight census variables, was associated with PTB in four US states between 1995–2001 [24]. Likewise, living in CTs with high unemployment, low education, and high poverty was associated with increased odds of PTB among non-Hispanic white and non-Hispanic black women, with larger effect sizes among non-Hispanic white women [21]. A study from California of the black-white disparity in PTB found that SES, measured by both maternal level of education and CT poverty, contributed to disparate rates, especially for births less than 32 weeks' gestation [23]. Additional studies

have found associations between PTB and CT-level median household income in Louisiana (1997–1998) [25] and very high gentrification (percent change in education level, poverty level, and median household income) in New York City (2008–2010) [26]. In contrast, one study found no increased risk of PTB with CT-level median household income in Massachusetts (1996–2002) [27] and another using a composite variable for neighborhood-level SES derived from seven census variables found no association between low neighborhood SES and PTB among a sample of 6,390 US black women after adjustment for individual-level characteristics [28]. A meta-analysis of 42 studies found that the association between segregation and PTB differed by race, with segregation being associated with an increased odds of PTB primarily among non-Hispanic black women [22].

Further, a growing body of research has assessed area-based measures of SES in conjunction with environmental exposures to investigate PTB from an environmental justice perspective. Such studies have found that exposures to air pollution and water pollution increase the risk for PTB, particularly among neighborhoods of low SES [70–72]. Future studies should consider the effects of these potential explanatory factors on gestational age at birth outcomes for women according to race/ethnicity and geography.

In our analysis, the inclusion of additional covariates related to maternal health conditions and behaviors did not meaningfully affect the associations between maternal level of education or neighborhood-level measures of SES and gestational age at birth. The number of cohorts contributing information on the additional covariates of interest was limited, thereby reducing our statistical power. Our results suggest, however, that both individual- and neighborhood-level measures of SES exert their influence partly through pathways that are not directly related to health status and/or behaviors, which might include pathways mediated by psychosocial factors, social support and stress, environmental exposures, and access to health care, healthy food, and recreation [29, 73, 74].

## Strengths and limitations

This study addresses an important gap in the literature, namely the limited data on racial/ethnic and geographic variation in individual- and neighborhood-level measures of SES and gestational age at birth. A strength of the ECHO-wide cohort is its heterogeneity in race/ethnicity, geographic variation covering multiple states and rural/urban areas, and SES. Another strength of the ECHO-wide cohort data is the availability of maternal address at the cohort level, which allowed for geocoding and assignment of neighborhood-level measures of SES at the CT level. Publicly-available birth record data does not allow for discernment of geography to the level of CT. Within the published literature, there is not another study that has brought together extant data from multiple pediatric cohorts across multiple states to address whether individual- and neighborhood-level markers of socioeconomic status associate with gestational age at birth across the United States. Existing literature that has considered individual- and neighborhood-level markers of socioeconomic status have mostly focused within a single state or has been carried out at a geographic level larger than CT (such as city or county). An additional strength of the ECHO cohort data is the availability and wealth of data on maternal antecedent health conditions and pregnancy complications that are linked with gestational age at birth outcomes. This rich data available across the ECHO cohorts offer a unique opportunity to expand on the analyses of geographic variation in gestational age outcomes based on natality files. Conversely, this heterogeneity of the ECHO cohorts also presents limitations. There was considerable variability in the nature of measurement of the different variables, and thus there is the potential for misclassification of outcomes (particularly given the variability in the range of methods that could be used to classify gestational age at births across participating cohorts)

and exposures. A limitation related to the assignment of CT level variables via ACS is that we used a single 5-year file across all cohorts (the 2005–2009 five-year release) including for birth years before or after that period, which could have resulted in some misclassification given that there could have been temporal changes in the SES indicators in some areas that would not have been captured by using the 2005–2009 year period. For births before 2005–2009, there is not a corresponding five-year ACS file that contains estimates for every CT (2005–2009 is the first available release of this data). While there is a five-year release for data after that period, we chose the ACS 2005–2009 dataset because, during the analysis planning stage, 2008 seemed to be the approximate mid-point of participant enrollment. Furthermore, it was deemed infeasible for us to use multiple five-year data releases as this would have required complex revisions and additions to the DeGAUSS tool, dissemination of analytic code to the cohorts to run locally using time-weighted averages, as well as extensive training of local cohort personnel.

This analysis was a disseminated meta-analysis, undertaken among extant data available within participating ECHO cohorts in order to facilitate team science without the need to share data outside of a given cohort. Since cohorts with only births in one gestational age outcome category (e.g., preterm-only cohorts) could not participate in this disseminated meta-analysis, this resulted in an under-representation of PTBs in the extant ECHO data and also prohibited the subclassification of PTBs into subcategories, which are known to have different antecedents [75]. In addition, the analytical approach to the analysis including model specifications, selection of covariates to include in the models, and missing data approaches all needed to be specified prior to sending the statistical code to the cohorts. This did not allow for flexibility and changes in the analysis and analytical approaches. Small cohort-specific sample sizes rendered some estimates unstable and therefore could not be used in the meta-analysis and resulted in varying numbers of cohorts contributing to each exposure-outcome combination.

This study did not examine maternal life course exposure to neighborhood-level SES. Neighborhoods that a mother has lived in throughout her life, encompassing childhood, adolescence, and adulthood, may have significant effects on her health later in life. Living in a poorer neighborhood as a youth may expose a mother to chronic stress that later increases her risk of giving birth to a preterm baby [76]. A study of African American, white, and Latina births in California in 1982–2011 found that living in neighborhood poverty at early life and adult time points was associated with a mother being at higher risk of PTB compared to living in a high-opportunity neighborhood throughout the life course [77]. An additional limitation is that paternal level of education was not a variable that was widely available across the participating cohorts and was not included in this analysis. Paternal level of education has been independently associated with risk for PTB [78].

A final limitation of the present study is its inability to investigate the effect of racism on racial/ethnic disparities in gestational age at birth in the United States. This is an important limitation considering that individual- and neighborhood-level measures do not adequately account for observed disparities. A systematic review of the literature including 15 studies published between 2009 and 2015 observed that racial discrimination is a significant risk factor for delivering preterm [79]. A multi-state analysis using PRAMS data for 2004 through 2012 observed that for non-Hispanic black women, experiences of racism were significantly associated with greater odds of delivering preterm [15]. Maternal experiences of racism and discrimination may more directly affect a pregnant woman's stress levels, and ultimately gestational age at birth. These experiences may include microaggressions over a lifetime and long-term exposure to chronic stress over time. A study of African-American women in Detroit, Michigan, in 2009–2011 found that perceived racial microaggressions in the past year were associated with

higher risk of PTB [80]. Another study of non-Latino white and black births in California from 2011–2014 found that chronic worry about racial discrimination was associated with increased risk of PTB. After adjusting for chronic worry about racial discrimination, the disparity in PTB among black and white mothers was noticeably attenuated [81].

## Next steps and future directions

These data suggest that interventions that promote higher educational attainment among women of reproductive age could contribute to a reduction in PTB in the United States, particularly in the South and Midwest. Future ECHO studies will involve individual-level data analyses examining gestational age as a continuous variable across all ECHO cohorts. Individual-level analyses will allow preterm cohorts to participate, increasing PTB sample size and our ability to evaluate subcategories of PTB. Further, individual-level data analyses will enable the investigation of the likely complex interrelationships among maternal education, neighborhood-level factors and additional factors across the life course and gestational age at birth outcomes and how these vary according to maternal race/ethnicity and US geography.

## Supporting information

**S1 Table. Data dictionary.**
(DOCX)

**S2 Table. Race and ethnicity frequencies of mothers of singleton live births by gestational age at birth category.**
(DOCX)

**S3 Table. Maternal education, neighborhood-level socioeconomic status, and adjusted odds of preterm, early-term, and late-post-term births compared to full-term births overall and by maternal race/ethnicity.**
(DOCX)

**S4 Table. Maternal race and ethnicity (S4A), percentage of black population in the census tract above or below the national average (S4B), percentage of population below the federal poverty level in the census tract above or below the national average (S4C), and urban vs. rural within each US census region by gestational age at birth category.**
(XLSX)

**S1 Ethics. Institutional review boards providing review and approval of research of the participating cohorts.**
(DOCX)

**S1 Funding. Grant support for each of cohorts contributing aggregate data results.**
(DOCX)

## Acknowledgments

The authors wish to thank our ECHO colleagues, the medical, nursing and program staff, as well as the children and families participating in the ECHO cohorts. We also acknowledge the contribution of the following program collaborators for Environmental Influences on Child Health Outcomes:

**ECHO Components—Coordinating Center**: Duke Clinical Research Institute, Durham, North Carolina: Benjamin DK, Smith PB, Newby KL; **Data Analysis Center**: Johns Hopkins University Bloomberg School of Public Health, Baltimore, Maryland: Jacobson LP*; Research

Triangle Institute, Durham, North Carolina: Parker CB*; **Person-Reported Outcomes Core**: Northwestern University, Evanston, Illinois: Gershon R, Cella D; **Children's Health and Exposure Analysis Resource**: Icahn School of Medicine at Mount Sinai, New York City, New York: Teitelbaum S; Wright RO; Wadsworth Center, Albany, New York: Aldous, KM, RTI International, Research Triangle Park, North Carolina: Fennell T; University of Minnesota, Minneapolis, Minnesota: Hecht SS, Peterson L; Westat, Inc., Rockville, Maryland: O'Brien B; **IDeA States Pediatric Trials Network**: University of Arkansas for Medical Sciences, Little Rock: Lee JY, Snowden J.

* Lead authors for the program collaborators for Environmental Influences on Child Health outcomes: Lisa P. Jacobson (ljacobs1@jhu.edu) and Corette B. Parker (rette@rti.org).

**ECHO Awardees and Cohorts that contributed aggregate data results for this analysis**:.

| Cohort ID | Cohort Name | Cohort Contact PI |
|---|---|---|
| 10101 | ECHO in Puerto Rico | Akram Alshawabkeh |
| 10401 | 35th Multicenter Airway Research Collaboration | Carlos Camargo |
| 10402 | 43rd Multicenter Airway Research Collaboration | Carlos Camargo |
| 10601 | Healthy Start | Dana Dabelea |
| 10801 | Boricua Youth Study | Cristiane Duarte |
| 10901 | Atlanta ECHO Cohort of Emory University | Anne Dunlop |
| 11001 | Safe Passage Study | Amy Elliott |
| 11201 | PETALS | Assiamira Ferrara |
| 11202 | KPRB | Assiamira Ferrara |
| 11303 | Tucson Children's Respiratory Study | James Gern |
| 11304 | Tucson Infant Immune Study | James Gern |
| 11305 | Wisconsin Infant Study Cohort | James Gern |
| 11306 | Childhood Origins of Asthma Study | James Gern |
| 11307 | Urban Environment and Childhood Asthma | James Gern |
| 11309 | Infant Susceptibility to Pulmonary Infections and Asthma Following RSV Exposure | James Gern |
| 11310 | Epidemiology of Home Allergens and Asthma Study | James Gern |
| 11311 | Wayne County Health Environment Allergy and Asthma | James Gern |
| 11312 | Childhood Allergy/Asthma Study | James Gern |
| 11401 | MADRES | Frank Gilliland |
| 11601 | ReCHARGE: Revisiting CHildhood Autism Risks from Genes and the Environment Study | Irva Hertz-Picciotto |
| 11701 | Pittsburgh Girls Study | Alison Hipwell |
| 11801 | New Hampshire Birth Cohort Study | Margaret Karagas |
| 11901 | CANDLE | Catherine Karr |
| 11902 | The Infant Development and Environment II Study | Catherine Karr |
| 11903 | GAPPS | Catherine Karr |
| 12101 | Early Growth and Development Study | Leslie Leve |
| 12102 | Early Growth and Development Study—Cohort II | Leslie Leve |
| 12103 | Early Parenting of Children | Leslie Leve |
| 12201 | Understanding Risk Gradients from Environment on Native American Child Health Trajectories: Toxicants, Immunomodulation, Metabolic syndromes, & Metals Exposure | Johnnye Lewis |
| 12301 | VDAART | Augusto Litonjua |
| 12401 | Vitamin C to Decrease Effects of Smoking in Pregnancy on Infant Lung Function | Cynthia McEvoy |
| 12402 | In-Utero Smoke, Vitamin C, and Newborn Lung Function | Cynthia McEvoy |

(*Continued*)

(Continued)

| | | |
|---|---|---|
| 12501 | Kennedy Krieger—Baby Siblings Research Consortium | Craig Newschaffer |
| 12502 | University of California Davis—Baby Siblings Research Consortium | Craig Newschaffer |
| 12503 | Autism Spectrum Disorders-Enriched Risk—University of Washington—BRSC | Craig Newschaffer |
| 12504 | Autism Spectrum Disorders-Enriched Risk-BRSC University of Miami | Craig Newschaffer |
| 12505 | University of Washington-Infant Brain Imaging Study | Stephen Dager |
| 12506 | Autism Spectrum Disorders—Enriched Risk—IBIS—University of Washington, St. Louis | Craig Newschaffer |
| 12507 | Infant Brain Imaging Study | Robert Schultz |
| 12508 | Infant Brain Imaging Study | Joseph Piven |
| 12509 | Early Autism Risk Longitudinal Investigation | Heather Volk |
| 12510 | Early Autism Risk Longitudinal Investigation | Rebecca Schmidt |
| 12511 | Early Autism Risk Longitudinal Investigation | Lisa Croen |
| 12512 | Autism Spectrum Disorders—Enriched Risk EARLI—Drexel University | Craig Newschaffer |
| 12513 | University of California—Markers of Autism Risk in Babies | Craig Newschaffer |
| 12601 | Rochester | Tom O'Connor |
| 12701 | Project Viva | Emily Oken |
| 12901 | ARCH | Nigel Paneth |
| 13001 | Mothers and Newborns | Frederica Perera |
| 13101 | Illinois Kids Development Study | Susan Schantz |
| 13102 | Chemicals in our Bodies | Susan Schantz |
| 13201 | Utah's Children's Project | Joseph Stanford |
| 13301 | Safe Passage Study | Leonardo Trasande |
| 13501 | Asthma Coalition on Community, Environment & Stress | Rosalind Wright |
| 13502 | Programming of Intergenerational Stress Mechanisms | Rosalind Wright |
| 13503 | Inova Childhood Longitudinal Study | Rosalind Wright |

The content is solely the responsibility of the authors and does not necessarily represent the official views of the National Institutes of Health.

## Author Contributions

**Conceptualization:** Anne L. Dunlop, Lyndsay Alvalos, Cristiane Duarte, Raina Fichorova, James Gern, Kathi Huddleston, Margaret R. Karagas, Ken Kleinman, Cynthia McEvoy, Nigel Paneth, Sheela Sathyanarayana, Jessica Snowden, Joseph B. Stanford, William Wheaton, Rosalind J. Wright, Monica McGrath.

**Data curation:** Ximin Li, Yijun Li.

**Formal analysis:** Ximin Li, Yijun Li, Monica McGrath.

**Investigation:** Anne L. Dunlop, Alicynne Glazier Essalmi, Lyndsay Alvalos, Carrie Breton, Carlos A. Camargo, Whitney J. Cowell, Dana Dabelea, Stephen R. Dager, Cristiane Duarte, Amy Elliott, Raina Fichorova, James Gern, Monique M. Hedderson, Kathi Huddleston, Margaret R. Karagas, Leslie Leve, Augusto Litonjua, Yunin Ludena-Rodriguez, Juliette C. Madan, Julio Mateus Nino, Cynthia McEvoy, Thomas G. O'Connor, Amy M. Padula, Nigel Paneth, Frederica Perera, Sheela Sathyanarayana, Rebecca J. Schmidt, Robert T. Schultz, Jessica Snowden, Joseph B. Stanford, Leonardo Trasande, Heather E. Volk.

**Methodology:** Ken Kleinman, William Wheaton, Rosalind J. Wright, Monica McGrath.

**Writing – original draft:** Anne L. Dunlop, Alicynne Glazier Essalmi, Lyndsay Alvalos, Carrie Breton, Carlos A. Camargo, Dana Dabelea, Cristiane Duarte, Raina Fichorova, Monique M.

Hedderson, Elizabeth Hom Thepaksorn, Margaret R. Karagas, Ken Kleinman, Ximin Li, Yijun Li, Juliette C. Madan, Julio Mateus Nino, Cynthia McEvoy, Amy M. Padula, Nigel Paneth, Sheela Sathyanarayana, Jessica Snowden, Monica McGrath.

**Writing – review & editing:** Anne L. Dunlop, Alicynne Glazier Essalmi, Lyndsay Alvalos, Carrie Breton, Carlos A. Camargo, Whitney J. Cowell, Dana Dabelea, Stephen R. Dager, Cristiane Duarte, Amy Elliott, Raina Fichorova, James Gern, Monique M. Hedderson, Elizabeth Hom Thepaksorn, Kathi Huddleston, Margaret R. Karagas, Ken Kleinman, Leslie Leve, Augusto Litonjua, Yunin Ludena-Rodriguez, Juliette C. Madan, Julio Mateus Nino, Cynthia McEvoy, Thomas G. O'Connor, Amy M. Padula, Nigel Paneth, Frederica Perera, Sheela Sathyanarayana, Rebecca J. Schmidt, Robert T. Schultz, Jessica Snowden, Joseph B. Stanford, Leonardo Trasande, Heather E. Volk, William Wheaton, Rosalind J. Wright, Monica McGrath.

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
