## [Decision Letter · Decision Letter 0]

9 Jun 2020

PONE-D-20-14238

Racial and Geographic Variation in Effects of Maternal Education and Neighborhood-level Measures of Socioeconomic Status on Gestational Age at Birth: Findings from The ECHO Cohorts

PLOS ONE

Dear Dr. Dunlop,

Thank you for submitting your manuscript to PLOS ONE. After careful consideration, we feel that it has merit but does not fully meet PLOS ONE’s publication criteria as it currently stands. Therefore, we invite you to submit a revised version of the manuscript that addresses the points raised during the review process.

We look forward to receiving your revised manuscript.

Kind regards,

Kelli K Ryckman

Academic Editor

PLOS ONE

Journal Requirements:

"No; the funders had no role in study design, data collection and analysis, decision to publish, or preparation of the manuscript."

4. One of the noted authors is a group or consortium [Environmental Influences on Child Health Outcomes]. In addition to naming the author group, please list the individual authors and affiliations within this group in the acknowledgments section of your manuscript. Please also indicate clearly a lead author for this group along with a contact email address.

6. We note that Figure 1 in your submission contain [map/satellite] images which may be copyrighted. All PLOS content is published under the Creative Commons Attribution License (CC BY 4.0), which means that the manuscript, images, and Supporting Information files will be freely available online, and any third party is permitted to access, download, copy, distribute, and use these materials in any way, even commercially, with proper attribution. For these reasons, we cannot publish previously copyrighted maps or satellite images created using proprietary data, such as Google software (Google Maps, Street View, and Earth). For more information, see our copyright guidelines: http://journals.plos.org/plosone/s/licenses-and-copyright.

6.1.    You may seek permission from the original copyright holder of Figure 1 to publish the content specifically under the CC BY 4.0 license.

6.2.    If you are unable to obtain permission from the original copyright holder to publish these figures under the CC BY 4.0 license or if the copyright holder’s requirements are incompatible with the CC BY 4.0 license, please either i) remove the figure or ii) supply a replacement figure that complies with the CC BY 4.0 license. Please check copyright information on all replacement figures and update the figure caption with source information. If applicable, please specify in the figure caption text when a figure is similar but not identical to the original image and is therefore for illustrative purposes only.

Reviewers' comments:

Reviewer's Responses to Questions

**Comments to the Author**

1. Is the manuscript technically sound, and do the data support the conclusions?

Reviewer #1: Yes

Reviewer #2: Partly

2. Has the statistical analysis been performed appropriately and rigorously? 

Reviewer #1: Yes

Reviewer #2: No

3. Have the authors made all data underlying the findings in their manuscript fully available?

Reviewer #1: No

Reviewer #2: Yes

4. Is the manuscript presented in an intelligible fashion and written in standard English?

Reviewer #1: Yes

Reviewer #2: Yes

5. Review Comments to the Author

Reviewer #1: This article examines differences in preterm birth risk by maternal individual-level (education) and neighbourhood-level (various census tract indicators), race/ethnicity and geographic region in ECHO Cohorts. Maternal education emerged as a predictor of preterm birth risk, and additional findings were detected by race/ethnicity, geographic region, and neighbourhood-level SES indicators. Overall, this is a very clear and well-written paper that uses a novel participant pool (ECHO cohorts) to address an important question. Specific comments are below.

Could the authors provide more information on ECHO cohorts and how a specific cohort could be included? Given the epidemiology nature of these analyses, the focus on broad geographic regions, and that more than 20 cohorts are represented, I was surprised that the sample size was relatively small (~ 25,000). For example, I am aware of epidemiology cohorts that are representative of wide geographic regions and that would meet the author’s inclusion criteria and that have at least 1,000,000 or more participants each. Given that sample size was cited as a potential limitation and reason for why some effects were not detected, it would be valuable for the authors to provide more context for the ECHO project, and the novel use of ECHO project data relative to this question specifically.

The authors focused on maternal education as an indicator of individual-level SES. Could the authors comment on whether they would expect different or similar patterns of findings if other SES indicators were available? For example, one strategy is to get mother and partner SES indicators, and take the highest as the indicator (even if mothers have low education, the status of their more-educated partner could increase their status by proxy), but only mother education level is available here. Other studies focus on income and occupation status as individual-level indicators. Given that the authors discuss how the benefits of education can vary by race/ethnicity, e.g. lower salary despite equivalent education attainment for Black individuals, it would be valuable to comment on this.

The census-level tract analyses were interesting. Could the authors provide more justification or context for their use of those particular indicators? For example, I thought that percent Black population per census tract was an interesting choice, especially given that non-Black, non-White races/ethnicities were also considered. Is percent Black individuals a particularly valuable indicator of neighbourhood-level SES for other non-White races/ethnicities, or would percent of other races/ethnicities also potentially matter?

Once criticism that I have seen of using a categorical approach to education is that racial/ethnic distributions within each category might not be equivalent. For example, when grouping bachelors or higher individuals, even within that category White individuals could be higher SES (e.g. more likely to have post-graduate training) compared to Black individuals (more likely to have a bachelor’s degree). Using continuous variables could reduce the risk of this potential confound. Why did the authors chose to use categorical rather than continuous variables? And why were these specific categories chosen?

The authors used a stratification approach to look at differences in the association between education and gestational length by race/ethnicity, geographic region, etc. Did the authors test any interaction terms or conduct any omnibus tests to determine if, overall, these differences in groups or moderating effects were statistically significant?

Given that differences did emerge for the Non-Hispanic Other Race category, could the authors provide a detailed racial/ethnic breakdown of this group in the supplemental materials? Although the focus has traditionally been on Black and Hispanic racial/ethnic groups, this information could be helpful for researchers examining health disparities in other racial/ethnic groups.

There were some interesting findings by geographic region and rurality of residence, but my first thought was how much overlap is there in these constructs? Could neighbourhood SES indicators be conflated by geographic region? It would be helpful if the authors included breakdowns of race/ethnicity and neighbourhood-level SES indicators by geographic region, at least in supplemental materials. Or if the authors included other SES indicators as covariates in regional analyses, that should be clearly stated in conclusions.

Could the authors provide more comment on the regional differences detected in the discussion? For example, are there regional differences in terms of laws or regulations that affect access to health care or education? Are there differences in societal or governmental wealth between these regions, or urban verses rural? And could these factors be relevant? The authors note that college education is rarer in the Midwest and South – are they suggesting that, as a consequence, individuals who do attain a college education in these regions are comparatively more privileged and affluent than similarly educated people in the West or Northeast?

Reviewer #2: This manuscript examined the association between maternal education (individual-level) and neighborhood level SES (urbanicity, %black, %poverty at census tract level) and gestational age at birth. My main concerns include the use of census tract instead of census block groups to examine neighborhood SES, the lack of use on residential history, and the use of only a single 5-year ACS estimates.

1. Introduction: There are many existing studies focusing on SES and preterm delivery. More details are needed to explain what added values this study brings to the field and what limitations from previous studies this study can address.

2. Geocoding: is residential history during pregnancy available? Why only using the earliest address collected during pregnancy?

3. Neighborhood-level SES variables: census tract is not ideal to assess SES as numerous findings have shown that there can be large heterogenities in SES within a census tract. Please use census block group instead.

4. Line 225: why only 2005-2009 ACS was used? The selection of specific ACS data should correspond to women's pregnancy period: e.g., you can use the middle year of the 5-year ACS as the index year, and generate time-weighted averages for each woman.

5. Lines 225-226: it is unclear why only three varaibles were used. Several well-established SES indices such as the Neighborhood Deprivation Index or Area Deprivation Index can be used to characterize SES, which are usually derived based on dozens of variables.

6. Covariates: why income was not included? In addition, is there any information on gestation week when prenatal care started? The binary prenatal care status won't tell much as the majority of women received prenatal care. The covairates selection should be guided by DAG.

7. Statistical modeling: it is unclear why the authors run seperate analsyes on each indivdiual cohort and then pool the results together using a meta-analysis. This is usually done when people don't have direct access to the raw data from each site. The authors may confirm whether this is the case. Otherwise, the authors should use a mixed effects model including data from all sites with random intercepts by cohort and by census block group.

8. In addition to the multinomial logistic regression, the authors may also treat the gestational age as a continuous outcome and model it use linear regression.

9. Limitation: Another limitation is the potential selection bias. It is not clear how representative the study samples are. There is no site in TX and FL, the states with the second and third most population in the US.

6. PLOS authors have the option to publish the peer review history of their article (what does this mean?). If published, this will include your full peer review and any attached files.

Reviewer #1: No

Reviewer #2: No

---

## [Author Response · Author response to Decision Letter 0]

15 Oct 2020

Reviewer #1 Comments to the Author

1. Could the authors provide more information on ECHO cohorts and how a specific cohort could be included? Given the epidemiology nature of these analyses, the focus on broad geographic regions, and that more than 20 cohorts are represented, I was surprised that the sample size was relatively small (~ 25,000). For example, I am aware of epidemiology cohorts that are representative of wide geographic regions and that would meet the author’s inclusion criteria and that have at least 1,000,000 or more participants each. Given that sample size was cited as a potential limitation and reason for why some effects were not detected, it would be valuable for the authors to provide more context for the ECHO project, and the novel use of ECHO project data relative to this question specifically.

Response: In the Objective section of the Revised Manuscript (page 7), we have added more information to describe the ECHO consortium of cohorts and how they were selected (and reference an article that goes into more detail on this topic). We also added language to the Objective section (pages 7 & 8) to introduce that ECHO uses both extant data (in collective analyses, such as this) and a common protocol to collect new data. 

2. The authors focused on maternal education as an indicator of individual-level SES. Could the authors comment on whether they would expect different or similar patterns of findings if other SES indicators were available? For example, one strategy is to get mother and partner SES indicators, and take the highest as the indicator (even if mothers have low education, the status of their more-educated partner could increase their status by proxy), but only mother education level is available here. Other studies focus on income and occupation status as individual-level indicators. Given that the authors discuss how the benefits of education can vary by race/ethnicity, e.g. lower salary despite equivalent education attainment for Black individuals, it would be valuable to comment on this.

Response: In the Introduction section of the Revised Manuscript (page 6) we provided a statement indicating that the literature supports that across “various measures of individual-level SES (including maternal level of education and income, marital and employment status, and type of health insurance) there is a consistent social gradient in the risk of preterm birth.”

Author Response to Editor’s Requests

Response: We can amend our current ethics statement (SmartForm) to include the information below, which provides the name of each Institutional Review Board for the cohorts involved in this research. 

In the Methods section of the Revised Manuscript (page 10), we also added the following language: “We used the categorical variable of maternal level of education as the main individual-level measure of SES based on existing research among US populations that support that education is the dimension of SES that most strongly and consistently predicts health, especially for women and children (references 35, 36). Also, maternal level of education as a categorical variable was more available across the ECHO cohort extant data. Paternal level of education was less often collected by the cohorts or, when collected, was oftentimes missing. Similarly, household income was less available across cohorts in that either entire cohorts did not collect this data or the variable response was missing within cohorts. Also, in order to be meaningfully interpreted income in relation to the federal poverty line, household income must be combined with household density, which was also not uniformly available across cohorts and/or was missing within cohort extant data. Furthermore, even when using the federal poverty thresholds, the meaning of the various income groupings varies a great deal across states and according to rural/urban status.” 

3. One criticism that I have seen of using a categorical approach to education is that racial/ethnic distributions within each category might not be equivalent. For example, when grouping bachelors or higher individuals, even within that category White individuals could be higher SES (e.g. more likely to have post-graduate training) compared to Black individuals (more likely to have a bachelor’s degree). Using continuous variables could reduce the risk of this potential confound. Why did the authors chose to use categorical rather than continuous variables? And why were these specific categories chosen?

Response: See our Response to Reviewer #1, Comment #2, in which we address this issue (and page 10 of the Revised Manuscript). To summarize, the extant data on level of education available from the participating cohorts was collected data as a categorical variable, thus, categorical data is what was available across cohorts. The categorization of maternal education data reflected how cohorts collected and categorized the data, and is based on accepted categories of attainment (including those references in the manuscript that pertain to research within US populations).

4. The census-level tract analyses were interesting. Could the authors provide more justification or context for their use of those particular indicators? For example, I thought that percent Black population per census tract was an interesting choice, especially given that non-Black, non-White races/ethnicities were also considered. Is percent Black individuals a particularly valuable indicator of neighbourhood-level SES for other non-White races/ethnicities, or would percent of other races/ethnicities also potentially matter?

Response: In the Methods section of the Revised Manuscript (page 11), we have inserted parenthetical phrases indicating that we included the percentage black population in the census tract as a measure of segregation and the percentage population living below the poverty line as a measure of poverty. Also, in the Methods section of the Revised Manuscript (page 12) we have explained and referenced our rationale for selecting the single indicator measure of neighborhood-level SES rather than more complex indices. Specifically, we modeled our selection of census tract variables on the paper by Carmichael et al. (2017), who investigated the association between percent of the tract population with household income below the poverty level (as measure of neighborhood-level deprivation) and percent of the tract population that was black (as a measure of segregation) and preterm birth; in their analysis, the Carmichael paper also created an index that incorporated eight census tract variables representing multiple aspects of socioeconomic level following methods of Messer et al. (2006). The Carmichael paper found that the single measure of tract poverty correlated well with the multi-measure index (r = 0.86) and resulted in the same conclusions. Therefore, we chose to focus on the single indicator measures, rather than on the more complex indices, as in Carmichael et al. (2017), Krieger et al. (2003 and 2005), and Rehkopf et al. (2006). 

5. The authors used a stratification approach to look at differences in the association between education and gestational length by race/ethnicity, geographic region, etc. Did the authors test any interaction terms or conduct any omnibus tests to determine if, overall, these differences in groups or moderating effects were statistically significant?

Response: We did not test the significance of interaction terms nor conduct omnibus tests to determine if, overall, the observed differences were statistically significant. An interaction between maternal education, gestational age and region was not possible as most participating cohorts contributed to one geographic region only. No individual-level data was available to the ECHO Data Analysis Center (DAC) so all analyses needed to be conducted successfully at each specific cohort, and then meta-analyzed at the DAC. In the Revised Manuscript, in the Objectives and Methods sections, we have emphasized that this analysis reflects a disseminated meta-analysis of extant data from the ECHO cohorts performed before individual-level data could be shared across cohorts. 

6. Given that differences did emerge for the Non-Hispanic Other Race category, could the authors provide a detailed racial/ethnic breakdown of this group in the supplemental materials? Although the focus has traditionally been on Black and Hispanic racial/ethnic groups, this information could be helpful for researchers examining health disparities in other racial/ethnic groups.

Response: In the Revised Manuscript we reference and include a Supplemental Table (S2; page 21), which is a detailed racial distribution by gestational age category, to include the following race categories: White, Black, Asian, Native Hawaiian or Other Pacific Islander, American Indian or Alaska Native, Multiple race, other race. We did not calculate race/ethnicity-specific effect estimates for these additional categories as our cohort-specific sample sizes were too small resulting in unstable estimates. We have also included information providing the ethnicity distribution by gestational age category. 

7. There were some interesting findings by geographic region and rurality of residence, but my first thought was how much overlap is there in these constructs? Could neighbourhood SES indicators be conflated by geographic region? It would be helpful if the authors included breakdowns of race/ethnicity and neighbourhood-level SES indicators by geographic region, at least in supplemental materials. Or if the authors included other SES indicators as covariates in regional analyses, that should be clearly stated in conclusions.

Response: In the Results section of the Revised Manuscript (page 25), we have included reference to a Supplemental Table (S4), which displays the distribution of race/ethnicity and neighborhood-level SES by geographic region. We did not include other SES indicators as covariates in regional analyses. To the Discussion section of the Revised Manuscript (page 26), we also noted that it is possible that neighborhood SES is conflated by geographic region. In the Discussion section, we also note that future ECHO analyses utilizing individual-level participant data will allow these complex relationships to be explored.

8. Could the authors provide more comment on the regional differences detected in the discussion? For example, are there regional differences in terms of laws or regulations that affect access to health care or education? Are there differences in societal or governmental wealth between these regions, or urban verses rural? And could these factors be relevant? The authors note that college education is rarer in the Midwest and South – are they suggesting that, as a consequence, individuals who do attain a college education in these regions are comparatively more privileged and affluent than similarly educated people in the West or Northeast?

Response: To the Discussion section of the Revised Manuscript (page 26), we have added a brief discussion and citations to full reports that discuss how the variation in state-level policies that relate to maternal and child health and health outcomes.

Reviewer #2 Comments to the Author

1. Introduction: There are many existing studies focusing on SES and preterm delivery. More details are needed to explain what added values this study brings to the field and what limitations from previous studies this study can address.

Response: In the Objective section of the Revised Manuscript (pages 7 & 8), we have added narrative to further emphasize the importance of combining extant data across ECHO cohorts to examine associations by varying US region as well as the innovativeness of using the collective analyses (disseminated meta-analysis) process to combine extant data across cohorts when individual-level data cannot be shared. The Discussion section (Strengths and Limitations part) includes a discussion of the strengths and contributions of this literature to the existing literature, which may be better interpreted now that the Revised Manuscript better describes the ECHO cohorts and the use of extant data. 

2. Geocoding: is residential history during pregnancy available? Why only using the earliest address collected during pregnancy?

Response: Most of the cohorts participating in this disseminated meta-analysis of extant data did not collect residential history during pregnancy, but only collected the earliest address during pregnancy. We chose to use the earliest address during pregnancy because it was more available across participating cohorts. 

3. Neighborhood-level SES variables: census tract is not ideal to assess SES as numerous findings have shown that there can be large heterogenities in SES within a census tract. Please use census block group instead.

Response: In the Methods section of the Revised Manuscript (page 12), we provide an explanation and citations for our use of census tract level measures of SES. Briefly, we chose to base our neighborhood-level SES measures on census tract based on the existing literature that shows that among non-Hispanic Black, non-Hispanic White, and Hispanic men and women, measures of economic deprivation were most sensitive to expected socioeconomic gradients in health, with the most consistent results and maximal geocoding linkage evident for tract-level analyses. Furthermore, census tract–level analyses yielded the most consistent results with maximal geocoding linkage (i.e., the highest proportion of records both geocoded and linked to census-defined areas) (Krieger et al., 2003; Krieger et al., 2005). 

Furthermore, we chose to use DeGAUSS system for geocoding for the following reasons: it is a freely available distributed geocoding system that used the same nationwide underlying street address database (the Census Bureau’s Topologically Integrated Graphical Encoding and Referencing “TIGER” data); a single geocoding engine (embedded in DeGAUSS) resulting in all cohorts using the same underlying address matching code; and a single, uniform underlying street database (TIGER). A centralized DAC geocoding process was not possible since the DAC did not have access to individual-level data. Purchasing, installing, and executing a commercially-available geocoder at all cohort locations was not possible. 

4. Line 225: why only 2005-2009 ACS was used? The selection of specific ACS data should correspond to women's pregnancy period: e.g., you can use the middle year of the 5-year ACS as the index year, and generate time-weighted averages for each woman.

Response: To the Methods section of the Revised Manuscript (page 12) we also added narrative to describe why we chose to use the five-year ACS data for the period 2005-2009. 

5. Lines 225-226: it is unclear why only three variables were used. Several well-established SES indices such as the Neighborhood Deprivation Index or Area Deprivation Index can be used to characterize SES, which are usually derived based on dozens of variables.

Response: See our response to Reviewer #1, Comment #4. 

6. Covariates: why income was not included? In addition, is there any information on gestation week when prenatal care started? The binary prenatal care status won't tell much as the majority of women received prenatal care. 

Response: See our response to Reviewer #1, Comment #2. Also, across the extant data available within the cohorts, the gestational week at which prenatal care was initiated was not widely available and/or was not collected in a similar manner (by self-report vs. by medical record abstraction). 

7. Statistical modeling: it is unclear why the authors run separate analyses on each individual cohort and then pool the results together using a meta-analysis. This is usually done when people don't have direct access to the raw data from each site. The authors may confirm whether this is the case. Otherwise, the authors should use a mixed effects model including data from all sites with random intercepts by cohort and by census block group.

Response: In the Objective section of the Revised Manuscript (pages 7 and 8), we have added more information to describe the ECHO consortium’s plan to utilize collective analyses (disseminated meta-analysis) to combine data across cohorts for hypothesis-testing when identifiable participant-level data could not be shared. The authors did not have access to individual-level data, therefore each cohort performed cohort-specific analyses based on specifications provided to them by the DAC, and the DAC pooled the results together using a meta-analysis.

8. In addition to the multinomial logistic regression, the authors may also treat the gestational age as a continuous outcome and model it use linear regression.

Response: In the Discussion section of the Revised Manuscript, we did add a statement specifying that in the future ECHO studies will involve individual-level data analyses examining gestational age as a continuous variable (page 30). Because this study represents a disseminated meta-analysis in which no participant-level data was held by the Data Analysis Center (DAC), the DAC only received the point estimates and variance/co-variance matrix from each cohort’s own local analysis. The a priori analysis specified the use of categorical gestational age to estimate the associations. 

9. Limitation: Another limitation is the potential selection bias. It is not clear how representative the study samples are. There is no site in TX and FL, the states with the second and third most population in the US.

Response: See our response to Reviewer #1, Comment #1.

---

## [Decision Letter · Decision Letter 1]

16 Nov 2020

PONE-D-20-14238R1

Racial and Geographic Variation in Effects of Maternal Education and Neighborhood-level Measures of Socioeconomic Status on Gestational Age at Birth: Findings from The ECHO Cohorts

PLOS ONE

Dear Dr. Dunlop,

Thank you for submitting your manuscript to PLOS ONE. After careful consideration, we feel that it has merit but does not fully meet PLOS ONE’s publication criteria as it currently stands. Therefore, we invite you to submit a revised version of the manuscript that addresses the points raised during the review process.

We look forward to receiving your revised manuscript.

Kind regards,

Kelli K Ryckman

Academic Editor

PLOS ONE

Reviewers' comments:

Reviewer's Responses to Questions

**Comments to the Author**

1. If the authors have adequately addressed your comments raised in a previous round of review and you feel that this manuscript is now acceptable for publication, you may indicate that here to bypass the “Comments to the Author” section, enter your conflict of interest statement in the “Confidential to Editor” section, and submit your "Accept" recommendation.

Reviewer #1: All comments have been addressed

Reviewer #2: (No Response)

2. Is the manuscript technically sound, and do the data support the conclusions?

Reviewer #1: Yes

Reviewer #2: (No Response)

3. Has the statistical analysis been performed appropriately and rigorously? 

Reviewer #1: Yes

Reviewer #2: (No Response)

4. Have the authors made all data underlying the findings in their manuscript fully available?

Reviewer #1: No

Reviewer #2: (No Response)

5. Is the manuscript presented in an intelligible fashion and written in standard English?

Reviewer #1: Yes

Reviewer #2: (No Response)

6. Review Comments to the Author

Reviewer #1: This article examines differences in preterm birth risk by maternal individual-level (education) and neighbourhood-level (various census tract indicators), race/ethnicity and geographic region in ECHO Cohorts. Maternal education emerged as a predictor of preterm birth risk, and additional findings were detected by race/ethnicity, geographic region, and neighbourhood-level SES indicators. Overall, this is a very clear and well-written paper that uses a novel participant pool (ECHO cohorts) to address an important question. All my comments have been addressed.

Reviewer #2: The authors did not address my concerns very well. The current analyses did not fully leverage the potential of the ECHO study. The same analyses can be easily done using birth certificate data with larger sample sizes and minimal selection bias. Again, it is unclear what the added value this study brings to the field. The idea of collective analysis is not new. It has been widely used in other settings: e.g., pooled time-series analyses on air pollution studies. Below are my detailed comments.

1) Regarding the neighborhood SES variables: the authors stated that "census tract–level analyses yielded the most consistent results with maximal geocoding linkage (i.e., the highest proportion of records both geocoded and linked to censusdefined areas)". This is confusing. If the geocoding was performed using residential address, I don't understand why you can find the corresponding census tract but not the census block group.

2) The authors did not address my question why only the 2005-2009 ACS was used. ACS starts to provide 5-year estimate data since 2005-2009, and the latest data available is 2014-2018. There can be large temporal changes in the SES indicators in certain areas across the years, which cannot be addressed if only one 5-year estimate data is used.

3) The authors did not answer my question why only 3 variables were selected, and why other well-established SES indices were not used.

7. PLOS authors have the option to publish the peer review history of their article (what does this mean?). If published, this will include your full peer review and any attached files.

Reviewer #1: No

Reviewer #2: No

---

## [Author Response · Author response to Decision Letter 1]

20 Dec 2020

1. Reviewer 2: The current analyses did not fully leverage the potential of the ECHO study. The same analyses can be easily done using birth certificate data with larger sample sizes and minimal selection bias. It is unclear what the added value this study brings to the field. The idea of collective analysis is not new. It has been widely used in other settings: e.g., pooled time-series analyses on air pollution studies. 

Author Response: We respectfully disagree with this Reviewer comment for the following reasons: 

(1) The same analysis cannot easily be done with birth certificate data because in using publicly-available birth certificate data (available for all states via the National Center for Health Statistics) it is not possible to obtain a unit of geography smaller than county or zip code of maternal residence (and these are only for units with population > 100,000). Specifically, maternal census tract of residence (the geographic level of interest for this manuscript) is not publicly-available through National Center for Health Statistics, prohibiting the conduct of census-tract level analyses on a multi-stage large-scale basis with publicly available data, which we were able to accomplish with the ECHO data. While for a given state it would be possible to make a request to the state’s department of public health to obtain and enter into a data use agreement to use census tract-level geocodes for that state, this would be very time-consuming to accomplish across multiple states and, furthermore, many states simply do not share geocodes at the census tract-level. Thus, we believe that this analysis using census tract-level data available within the ECHO cohort makes a substantial contribution to the literature. 

(2) Within the published literature, there is not another disseminated meta-analysis that has brought together extant data from existing pediatric cohorts to address the research question of how individual- and neighborhood-level markers of socioeconomic status associate with gestational age at birth across the United States. Existing literature that has considered individual- and neighborhood-level markers of socioeconomic status have mostly focused in a single state or smaller geography of the United States, rather than including multiple regions across the United States, as this multi-cohort research does. 

In summary, we believe that this work does represent a unique manner of collaboration in which the existing pediatric cohorts came together through ECHO to perform a disseminated meta-analysis upon their extant data under the guidance of a central Data Analysis Center. Thus, both the findings of this research, around the role of individual- and neighborhood-level markers of socioeconomic status on gestational age at birth, as well as the methods of the use of data from multiple cohorts that do not yet have permission to share data, are of interest to readership. 

However, to address this important concern of the Reviewer, we have added a statement to the Discussion section of the R2 Manuscript (pages 28-29): “Another strength of the ECHO-wide cohort data is the availability of maternal address at the cohort level, which allowed for geocoding and assignment of neighborhood-level measures of SES at the CT level. Publicly-available birth record data does not allow for discernment of geography to the level of CT. Within the published literature, there is not another study that has brought together extant data from multiple pediatric cohorts or across multiple states to address whether individual- and neighborhood-level markers of socioeconomic status associate with gestational age at birth across the United States. Existing literature that has considered individual- and neighborhood-level markers of socioeconomic status have mostly focused within a single state or has been carried out at a geographic level larger than CT (such as city or county).”

2. Reviewer 2: Regarding the neighborhood SES variables: the authors stated that "census tract–level analyses yielded the most consistent results with maximal geocoding linkage (i.e., the highest proportion of records both geocoded and linked to census defined areas)". If the geocoding was performed using residential address, I don't understand why you can find the corresponding census tract but not the census block group.

Author Response: In our previous Response to Review and R1 manuscript, we included this statement: “We chose to consider SES measures at the level of census tract based on literature showing that among non-Hispanic black, non-Hispanic white and Hispanic men and women, measures of economic deprivation were most sensitive to socioeconomic gradients in health with the most consistent results and maximal geocoding evidence for census tract-level analyses (Krieger et al., 2003; Krieger et al., 2005).” This statement, about the maximal geocoding linkage, was referring to the experience of the authors of those manuscripts (not to our experience in the ECHO data). Also, this referenced statement was intended to provide rationale for why we chose to focus our analysis at the level of census tract (i.e., that literature supports that measures of economic deprivation at that level were most consistently associated with gradients in health outcomes).

To clarify any confusion and address this Reviewer concern further, we have edited this statement in the Methods section of the R2 Manuscript (pages 12-13) to more clearly convey those ideas: “We chose to consider SES measures at the level of CT based on literature showing that CT-level analyses resulted in maximal geocoding linkage (i.e., the highest proportion of records geocoded and linked to census-defined geography) and that measures of economic deprivation at the CT-level were most sensitive to expected socioeconomic gradients in health among non-Hispanic black, non-Hispanic white and Hispanic men and women (Krieger et al., 2003; Krieger et al., 2005).

3. The authors did not answer my question why only 3 variables were selected, and why other well-established SES indices were not used.

Author Response: In our previous Response to Review, we had replied to this question with the following narrative addition to the R1 Manuscript (page 12),: “We selected the three CT level markers of SES based on past research that has investigated the relationship between census tract markers of SES and preterm birth, which found that the single-measure variable of percentage of persons below poverty performed as well as more complex, composite measures of economic deprivation [23] in conjunction with our a priori interest in exploring census tract markers of rural/urban status and segregation in conjunction with poverty.” 

In the R2 Manuscript (page 12) we have added further detail to that previous response (the underlined stanza) to more thoroughly clarify our rationale for not using an established index: “We selected the three CT level markers of SES based on past research that investigated the relationship between census tract markers of SES and preterm birth, which found that the single-measure variable of percentage of persons below the poverty level performed as well as more complex, composite measures of economic deprivation [23] in conjunction with our a priori interest in exploring CT markers of rural/urban status and segregation in conjunction with poverty. Specifically, Carmichael et al. [23] found that the single-measure of percent of the CT population below the poverty level correlated well with an eight-measure index of CT variables (r=0.86) and resulted in the same conclusions.”

4. The authors did not address my question why only the 2005-2009 ACS was used. ACS starts to provide 5-year estimate data since 2005-2009, and the latest data available is 2014-2018. There can be large temporal changes in the SES indicators in certain areas across the years, which cannot be addressed if only one 5-year estimate data is used.

Author Response: In our previous Response to Review we did reply to this question by including a narrative response in the R1 manuscript (page 13): “We used five-year data releases from ACS as only the five-year releases are guaranteed to have estimates for every CT in the United States. In this disseminated meta-analysis, addresses were geocoded at the cohort level using DeGAUSS since participant address could not be shared with the DAC; disseminating analytical code to the cohorts to run locally using time-weighted averages for each woman was not possible across all of the participating cohorts.” 

We acknowledge, however, that this response did not completely capture the concern of the Reviewer. As such, in the R2 Manuscript, we have done the following:

a. To the Methods Section of the R2 Manuscript (pages 10-11), we have added further detail to our rationale for using the DeGAUSS tool for the disseminated geocoding across multiple cohorts and specifically detailed why we choose the 2005-2005 ACS 5-year release (pages 10-11), which now reads as follows: “DeGAUSS, a freely available distributed geocoding system, facilitates reproducible geocoding and geomarker assessment in multi-site studies while maintaining the confidentiality of protected health information by enabling the sites access to a single embedded geocoding engine that uses the same nationwide street address data base (the Census Bureau’s Topologically Integrated Graphical Encoding and Referencing, TIGER, data) and matching code without having to share address data. The DeGAUSS geocoder used current (at the time of the study) TIGER street centerline address range files to convert residential addresses into geographical coordinates and provided assessment of qualitative precision for the geocodes [(37)]. Using the latest TIGER street centerline address range files was appropriate because of improvements to street and address ranges made by census bureau over time. After geocoding, the resulting latitude/longitude coordinates were joined to the TIGER 2000 Census Tract boundaries. Since census tracts are revised only every ten years, the TIGER 2000 Census Tract boundaries were needed in order to correctly match the census tract for geocoded addresses to the 2005-2009 American Community Survey (ACS) 5-year estimates, which used the same census tract identifiers as the 2000 TIGER census tract layer. We chose the ACS 2005-2009 dataset because for births prior to 2005-2009 there is not a corresponding five-year ACS file that contains estimates for every CT (2005-2009 is the first available release of this data) and, during the analysis planning stage, 2008 seemed to be the approximate mid-point of participant enrollment.”

b. To the Discussion/Limitations Section of the R2 Manuscript (page 29) we have added text acknowledging that a limitation of our approach is the use of a single 5-year estimate across all births for a given cohort: “A limitation related to the assignment of CT level variables via ACS is that we used a single 5-year file across all cohorts (the 2005-2009 five-year release) including for birth years before or after that period, which could have resulted in some misclassification given that there could have been temporal changes in the SES indicators in some areas that would not have been captured by using the 2005-2009 year period. For births before 2005-2009, there is not a corresponding five-year ACS file that contains estimates for every CT (2005-2009 is the first available release of this data). While there is a five-year release for data after that period, we chose the ACS 2005-2009 dataset because, during the analysis planning stage, 2008 seemed to be the approximate mid-point of participant enrollment. Furthermore, it was deemed infeasible for us to use multiple five-year data releases as this would have required complex revisions and additions to the DeGAUSS tool, dissemination of analytic code to the cohorts to run locally using time-weighted averages, as well as extensive training of local cohort personnel.”

We believe that the strengths of the study far outweigh this limitation and the work represents a meaningful contribution to the literature in that within the published literature, there is not another disseminated meta-analysis that has brought together extant data from existing pediatric cohorts to address the research question of how individual- and neighborhood-level markers of socioeconomic status associate with gestational age at birth across the United States. Existing literature that has considered individual- and neighborhood-level markers of socioeconomic status have mostly focused in a single state or smaller geography of the United States, rather than including multiple regions across the United States, as this research does. 

Thank you for this opportunity to respond to the review.

---

## [Editor Report · Decision Letter 2]

22 Dec 2020

Racial and Geographic Variation in Effects of Maternal Education and Neighborhood-level Measures of Socioeconomic Status on Gestational Age at Birth: Findings from The ECHO Cohorts

PONE-D-20-14238R2

Dear Dr. Dunlop,

We’re pleased to inform you that your manuscript has been judged scientifically suitable for publication and will be formally accepted for publication once it meets all outstanding technical requirements.

Kind regards,

Kelli K Ryckman

Academic Editor

PLOS ONE

Additional Editor Comments (optional):

I have carefully reviewed the authors responses to reviewer #2's concerns and determined that the authors have done an excellent job of adequately addressing all concerns raised.
---

## [Editor Report · Acceptance letter]

28 Dec 2020

PONE-D-20-14238R2 

Racial and geographic variation in effects of maternal education and neighborhood-level measures of socioeconomic status on gestational age at birth: Findings from the ECHO cohorts 

Dear Dr. Dunlop:

I'm pleased to inform you that your manuscript has been deemed suitable for publication in PLOS ONE. Congratulations! Your manuscript is now with our production department. 

Kind regards, 

on behalf of

Dr. Kelli K Ryckman 

Academic Editor

PLOS ONE